# The unexplored diversity of rough-seeded lupins provides rich genomic resources and insights into lupin evolution

Karolina Susek [1,10] ✉, Leonardo Vincenzi [2,10], Magdalena Tomaszewska [1], Magdalena Kroc[1], Edoardo Franco [2], Emanuela Cosentino[3], Antonina Rita Limongi [2], Umesh Kumar Tanwar[1], Humaira Jamil [1], Matthew Nicholas Nelson [4], Philipp E. Bayer[5,6], David Edwards [7], Roberto Papa [8], Massimo Delledonne [2,3] & Scott A. Jackson [9]

Lupin crops provide nutritious seeds as an excellent source of dietary protein. However, extensive genomic resources are needed for crop improvement, focusing on key traits such as nutritional value and climate resiliency, to ensure global food security based on sustainable and healthy diets for all. Such resources can be derived either from related lupin species or crop wild relatives, which represent a large and untapped source of genetic variation for crop improvement. Here, we report genome assemblies of the cross-compatible species *Lupinus cosentinii* (Mediterranean) and its pan-Saharan wild relative *L. digitatus*, which are well adapted to drought-prone environments and partially domesticated. We show that both species are tetraploids, and their repetitive DNA content differs considerably from that of the main lupin crops *L. angustifolius* and *L. albus*. We present the complex evolutionary process within the rough-seeded lupins as a species-based model involving polyploidization and rediploidization. Our data also provide the foundation for a systematic analysis of genomic diversity among lupin species to promote their exploitation for crop improvement and sustainable agriculture.

The genus *Lupinus* (lupins) is part of the highly diverse legume family (Fabaceae), which has undergone spectacular evolutionary radiation[1]. Some species, such as *Lupinus albus*, *L. angustifolius*, *L. luteus* and *L. mutabilis*, are economically important animal feed crops that are particularly resilient to drought[2]. Others, such as *L. albus* and *L. mutabilis*, are considered as orphan crops dedicated to food production via the cultivation of traditional varieties and landraces in restricted geographical areas. *L. cosentinii* and *L. digitatus* are part-domesticated species associated with drought tolerance, and others

have the potential for agricultural exploitation, including *L. atlanticus* and *L. pilosus*[3,4]. Lupins have attracted interest due to their nutritional seeds[5] and potential for sustainable farming[6,7]. Lupin seeds can contribute to a healthy human diet[8,9] due to their protein content of up to 40%[10,11], and accordingly they are considered an important component of local and global food security[12]. The development of genomic tools could facilitate pre-breeding and breeding processes by exploiting the rich diversity of wild and domesticated lupin species[13], which we describe here as crop wild

[1]Legume Genomics Team, Institute of Plant Genetics, Polish Academy of Sciences, Poznan, Poland. [2]Functional Genomics Lab, Department of Biotechnology, University of Verona, Verona, Italy. [3]Genartis srl, Via Albere 17, 37138 Verona, Italy. [4]Floreat Laboratories, The Commonwealth Scientific and Industrial Research Organisation, Floreat, WA, Australia. [5]OceanOmics, The Minderoo Foundation, Perth, WA, Australia. [6]The UWA Oceans Institute, The University of Western Australia, Crawley, WA, Australia. [7]Centre for Applied Bioinformatics and School of Biological Sciences, University of Western Australia, Perth, WA, Australia. [8]Dipartimento di Scienze Agrarie, Alimentari e Ambientali, Università Politecnica delle Marche, Ancona, Italy. [9]Institute for Plant Breeding and Genetics, University of Georgia, Athens, GA, USA. [10]These authors contributed equally: Karolina Susek, Leonardo Vincenzi. ✉e-mail: ksus@igr.poznan.pl

relatives (CWRs), meaning wild species or weedy plants that are taxonomically related to domesticated lupins or can be used in agriculture (e.g. de novo domestication). However, to fully exploit genomics for crop improvement, the number of whole-genome and pangenome sequences available for legume crops must increase, including not only major domesticated gene pools but also wild species as well minor crops of the primary, secondary and tertiary gene pool[14,15]. The following whole-genome sequences are available for legume CWRs: peanut (*Arachis hypogaea*), *A. duranensis* and *A. ipaensis*[16]; soybean (*Glycine max*) and *G. soja*[17]; mung bean (*Vigna radiata* var. *radiata*, *V. reflexo-pilosa* var. *glabra* and *V. radiata* var. *sublobata*[18]; and a chickpea (*Cicer arietinum*) super-pangenome including wild species such as *C. reticulatum*, *C. judaicum* and *C. pinnatifidium*[19,20].

Whole-genome sequences have been published for three lupin crops, namely *L. angustifolius*[21,22], *L. albus*[23,24] and most recently *L. mutabilis*[25], providing insight into key aspects of lupin genome structure, diversity and evolution. However, information from CWRs is needed to take full advantage of lupin genetic resources. The genetic diversity of lupins has been highlighted by studies of chromosome number and genome size[26,27], as well as epigenomic[28] and phylogenetic analysis[29], and more recently the development of pangenomes for *L. albus*[30] and *L. angustifolius*[22].

There are ~275 lupin species conventionally divided into New World and Old World types, reflecting two main geographical centers of species diversity[4]. The distribution of Old World lupins has resulted from both climate change and human activities beginning in the Pleistocene epoch, whereas evolution within New World lupins has been enhanced by processes such as ecological differentiation and intensive hybridization. The main changes in the gene pool of wild populations may reflect disruptive differentiation caused by occasional hybridization and subsequent intergradations with escaped or neglected sporadically domesticated strains[31]. Human activity in Europe may have influenced the distribution of *L. angustifolius*[32]. Most annual and perennial lupins belong to the New World group and are found mainly in North America and the Andes, but only one species has been domesticated (*L. mutabilis*, 2n = 48). The Old World lupins comprise ~15 annual species distributed around the Mediterranean basin as well as North and East Africa[4], some of which are smooth-seeded and others rough-seeded species. The smooth-seeded species *L. albus*, *L. angustifolius* and *L. luteus* have been domesticated, along with the rough-seeded species *L. cosentinii*[4]. The domestication of lupins involved the introduction of desirable traits such as permeable seeds, non-shattering pods, early flowering and low alkaloid levels in seeds. *L. mutabilis* also has typical traits of the classical legume domestication syndrome, but the seed alkaloid content remains high in domesticated lines[33].

The somatic chromosome number of Old Word lupins varies widely (2n = 32, 36, 38, 40, 42, 50 or 52), with a basic chromosome number of x = 5–13. However, the highest chromosome numbers (2n = 40–52) tend to be found in the heterogeneous smooth-seeded group, whereas the morphologically and genetically more homogeneous rough-seeded species have fewer chromosomes (2n = 32–42)[26]. In contrast, most New World lupins have a somatic chromosome number of 2n = 36 or 48, the exceptions being *L. bracteolaris*, *L. linearis* (both 2n = 32 or 34), *L. cumulicola* and *L. villosus* (both 2n = 52), but the basic chromosome number is proposed to be x = 6 in all cases[34,35]. Multiple chromosome rearrangements have occurred among the Old World lupins, revealing a complex evolutionary process that suggests polyploidy[23,27,36]. There is evidence that *L. albus* and *L. angustifolius* evolved by genome duplication and/or triplication[21,23,36,37] from a diploid *Lupinus* ancestor[24]. Furthermore, processes such as aneuploidy may be unique to the Old World lupins[38,39], but aneuploid reduction from ancestral species has been reported in the legume family[38].

Lupins belong to the early-branching papilionoid genistoid clade[1], and are placed in the core genistoid clade. The estimated age of genistoid diversification is ~56 Ma[40], soon after the emergence of papilionoids ~58.6 Ma[40,41]. Numerous whole-genome duplication (WGD) events have been identified[42,43] following legume-common tetraploidy (LCT, ~59 Ma) and ancient core eudicot-common hexaploidy (ECH, ~130 Ma)[44]. Evidence of these common events remains in papilionoid species such as peanut[45] and soybean[44], highlighting their important impact in shaping legume genome structure and evolution. In papilionoids, one WGD has been shared by an entire subfamily, in the common ancestor of all papilionoids[42,43], even though several WGDs have occurred in papilionoids, as well as one whole-genome triplication (WGT) in the most recent common ancestor of Genisteae[42]. The genistoid clade shows the highest frequency of polyploidy but is poorly characterized and only weakly supported as a sister clade to the remaining core papilionoids (Doyle 2012). Early data suggest that the genistoid basic chromosome number was x = 9 and the most common somatic chromosome number was 2n = 18[43,46]. The WGT event has been identified in *L. albus* ~ 22 Ma[24] and *L. angustifolius* 20–30 Ma[37,43], indicating that the lupin diploid ancestor had a basic chromosome number of x = 9[24].

A more comprehensive view of polyploidization in legumes requires a denser sampling of taxa across the genistoid clade[43,47]. Old World lupin species: *L. cosentinii* Guss. (2n = 32) and *L. digitatus* Forsk. (2n = 36) are recognized to be drought tolerance, and *L. digitatus* serves as a source of drought-tolerance genes. *L. cosentinii* is native to the western Mediterranean coast but has been introduced in Austria, Romania, South Africa and, more recently in several parts of Australia[4,48]. Furthermore, *L. cosentinii* cv Erregulla was domesticated de novo in Australia from local wild germplasm in the 20th century, and has desirable traits such as soft, low-alkaloid seeds, non-shattering pods and early flowering[48]. In contrast, *L. digitatus* is native to the pan-Saharan region[4] and seeds of domesticated *L. digitatus* have been found in the tombs of Egyptian Pharaohs, suggesting domestication began >4000 years ago[49]. Interspecific crosses of rough-seeded lupins yielded the highest frequency of viable F₁ hybrids for *L. cosentinii*, *L. digitatus* and *L. atlanticus*, suggesting genomic similarity[4,50].

Here, we present high-quality genome assemblies for two rough-seeded lupin species (*L. cosentinii* and *L. digitatus*), revealing their genetic architecture and the consequences of polyploidy during lupin evolution. We also propose a model that will enable further studies within the *Lupinus* genus and the genistoid clade. Our comparative genomics analysis with other, smooth-seeded lupins provides insights into the complex evolutionary history of *Lupinus*, including potential rediploidization events following polyploidization.

## Results and discussion
### De novo genome assemblies of rough-seeded lupins
We used a combination of methods to generate genome assemblies of the rough-seeded lupin species *L. cosentinii* (2n = 32) and its wild rough-seeded relative *L. digitatus* (2n = 36). First, we produced PacBio HiFi reads (Supplementary Data 1, Supplementary Data 2, Supplementary Fig. 1) with ~55× coverage (~32.8 Gbp) for *L. cosentinii* and ~43× coverage (18.9 Gbp) for *L. digitatus*. We then used HiCanu to generate 650 and 492 Mbp assemblies for *L. cosentinii* and *L. digitatus*, respectively (Supplementary Data 3, Supplementary Data 4). The assemblies were polished using 39 and 37 Gbp of Illumina 150PE reads (Supplementary Data 1, Supplementary Data 2). Purging reduced the assembly size to 588 Mbp for *L. cosentinii* and 435 Mbp for *L. digitatus* (Supplementary Data 3, Supplementary Data 4). We applied two sequential approaches to scaffold the contigs, first with 560 Gbp (*L. cosentinii*) and 722 Gbp (*L. digitatus*) of Bionano optical maps (based on 4.7 million and 5.4 million molecules for *L. cosentinii* and *L. digitatus*, respectively), then with 60.4 Gbp (*L. cosentinii*) and 53 Gbp (*L. digitatus*) of chromosome-level Illumina Hi-C data (Supplementary Data 1,

**Table. 1 | Summary statistics of the final *Lupinus cosentinii*, *L. digitatus*, *L. albus* and *L. angustifolius* genome assemblies**

|  | *L. cosentinii* | *L. digitatus* | *L. albus* | *L. angustifolius* |
|---|---|---|---|---|
| Total assembly length (bp) | 588,329,261 | 435,543,761 | 450,972,408 | 653,266,117 |
| Total length of scaffolds (bp) (%) | 425,591,036 (72.3%) | 378,138,940 (86.8%) | 434,803,248 (96.4%) | 652,980,617 (99.95%) |
| Number of scaffolds | 19 | 22 | 26 | 20 |
| Scaffold N50 (bp) | 20,862,866 | 18,425,048 | 11,119,386 | 30,711,064 |
| Number of gaps | 36 | 72 | 52 | 571 |
| Gap size (bp) | 36,974 | 422,827 | 2,268,239 | 285,500 |
| Remaining contigs | 709 | 339 | 63 | / |
| Remaining contig total length (bp) | 162,738,225 | 57,404,821 | 16,169,160 | / |
| Remaining contig N50 (bp) | 491,921 | 335,629 | 397,328 | / |
| BUSCO analysis | C: 96.3% [S: 72.7%, D: 23.6%] | C: 95.3% [S: 72.8%, D: 22.5%] | C: 95.7% [S: 73.4%, D: 22.3%] | C: 96.7% [S: 73.9%, D: 22.8%] |

BUSCO statistics: C complete genes, S single-copy, D duplicates.

Supplementary Data 2, Supplementary Fig. 2). The resulting *L. cosentinii* genome (588 Mbp) had 19 scaffolds (~426 Mbp, ~72% of the assembled genome) and 709 further contigs, whereas the *L. digitatus* genome (435 Mbp) consisted of 22 scaffolds (~378 Mbp, ~87% of the assembled genome) and 339 remaining contigs (Table 1, Fig. 1). Benchmarking universal single-copy orthologs (BUSCO) reported 96.3% and 95.3% completeness along with 23.6% and 22.5% duplicated genes in *L. cosentinii* and *L. digitatus*, respectively. Both genomes were assembled at the highest possible level, including both Bionano optical maps and Hi-C data, but the high level of assembly duplication and genome ploidy hindered the reconstruction process, resulting in a more fragmented and less contiguous product than expected. This highlights the difficulties encountered when reconstructing complex genomes, such as those of plants, where even two of the most powerful genome scaffolding technologies can be ineffective. Lupin genomes that are already published (*L. albus* and *L. angustifolius*; Supplementary Fig. 3) were therefore used as references for comparison and downstream analysis. The *L. cosentinii* genome was larger than that of *L. digitatus*, which was similar to the reported 451-Mbp genome size of *L. albus*[23], but both genomes reported here were much smaller than the 653-Mbp genome of *L. angustifolius*[22]. The recently reported 620-Mbp *L. mutabilis* genome[25] has a BUSCO completeness of 94.8% with a duplication rate of 21.4% on the "Fabales" BUSCO database. Considering the BUSCO completeness in all five lupins, they each present a completeness level of ~96% and feature a similar proportion of duplicated genes. Our lupin genomes are similar in size to those of the common bean (*Phaseolus vulgaris*) and the model legumes *Medicago truncatula* and *Lotus japonicus*, which are ~580 Mbp[51], ~430 Mbp[52] and ~470 Mbp[53], respectively.

All lupin genomes based on whole-genome sequencing (WGS) were smaller than values estimated by flow cytometry because the latter technique is based on relative genome sizes (absolute sizes are more difficult to validate). Furthermore, the variable characteristics of plant tissues (e.g., abundance of secondary metabolites) and the use of different buffers, reagents and reference standards, can influence genome size determination[54]. Long-read sequencing therefore provides more precise estimates and can address inconsistencies, while also improving the accuracy of flow cytometry standards. Genome sizes based on *k*-mer analysis are usually smaller than those estimated by flow cytometry due to collapsed repeat regions as well as polyploidy[55,56].

## Gene structure and composition of repetitive sequences

An ab initio prediction supported by RNA-Seq data (22–32 samples, ~30 million Illumina 150PE fragments each) was used to annotate the two genomes. We predicted the presence of 34,780 and 31,260 genes in the *L. cosentinii* and *L. digitatus* genomes, respectively. For 26,860 (77.2%) and 25,478 (81.5%) of these genes, functional annotations were also present in high-confidence databases (SwissProt, RefSeq and

TAIR) (Supplementary Data 5 and 6) and in the proteomes of *L. albus* and *L. angustifolius* (Table 2). Functional annotation with Gene Ontology (GO) terms was possible for 23,544 *L. cosentinii* (67.7%) and 23,019 *L. digitatus* genes (73.6%). Repetitive DNA accounted for 352.5 Mbp (60%) and 206.3 Mbp (47.4%) of the *L. cosentinii* and *L. digitatus* genomes, respectively. The major classes of repetitive elements were simple repeats, representing 22.4% and 15.5% of the *L. cosentinii* and *L. digitatus* genomes, and long terminal repeat (LTR) retroelements, representing 21.3% and 20.1% of the *L. cosentinii* and *L. digitatus* genomes, respectively (Table 3, Fig. 2a). Both *L. albus* and *L. angustifolius* have a repetitive DNA content of 50–60%, compared to 64% for the recently characterized *L. mutabilis* genome, and have a much lower simple repeat content (~1%) than *L. cosentinii* and *L. digitatus* (~17%) but a higher content of LTR elements (~36% compared to ~21%).

Repetitive DNA and polyploidization are known to be key factors in the evolution of plant genomes[57], including lupins[7]. The repetitive DNA content of *L. cosentinii* and *L. digitatus* was comparable to the other lupins (50–60%), irrespective of species or ploidy, showing that no extensive amplification or reduction of repeat sequences caused significant variation in the overall content after speciation or polyploidization. However, both *L. cosentinii* and *L. digitatus* had a smaller portion of LTR elements than *L. albus* and *L. angustifolius*. Although LTR elements are the most prevalent transposable elements in plant genomes, the abundance of specific superfamilies can vary greatly between species and even within varieties of the same species. Several related species share the ability to amplify a superfamily[58], but LTR elements with a high copy number in one species may have a low copy number in a close relative[59]. Along with polyploidization, LTR elements may therefore be important for lupin genome evolution as suggested for the Fabaceae family more widely[60]. Indeed, retrotransposons and tandem repeats/microsatellites have influenced the evolution of *L. angustifolius*, and different processes such as repeat amplification, proliferation and clearance underlie the lineage-specific dynamics of repetitive sequences in lupins[61]. *L. cosentinii* and *L. digitatus* feature a higher number of simple repeats than *L. albus* and *L. angustifolius*, suggesting simple repeats played an important role in shaping the genome diversification in lupins. A large increase in the content of simple repeats was reported in the *L. angustifolius* pangenome relative to the reference genome[22]. Given the extensive haplotype variation for transposable elements in many species[62], a large number of accessions of the same lupin species should be sequenced to investigate the dynamic evolution of polyploid plant genomes.

We aligned the Illumina WGS data on the two assembled genomes, revealing 21,645 deletions and 72 insertions in *L. cosentinii* as well as 441 deletions and 1831 insertions in *L. digitatus* (Fig. 2b). We focused our analysis of these structural variations (SVs) in the two main families of transposable elements. *L. cosentinii* featured more deletions in LTR elements (3420) and DNA transposons (11,160) than *L. digitatus* (304

# Genome assemblies of two rough-seeded lupins

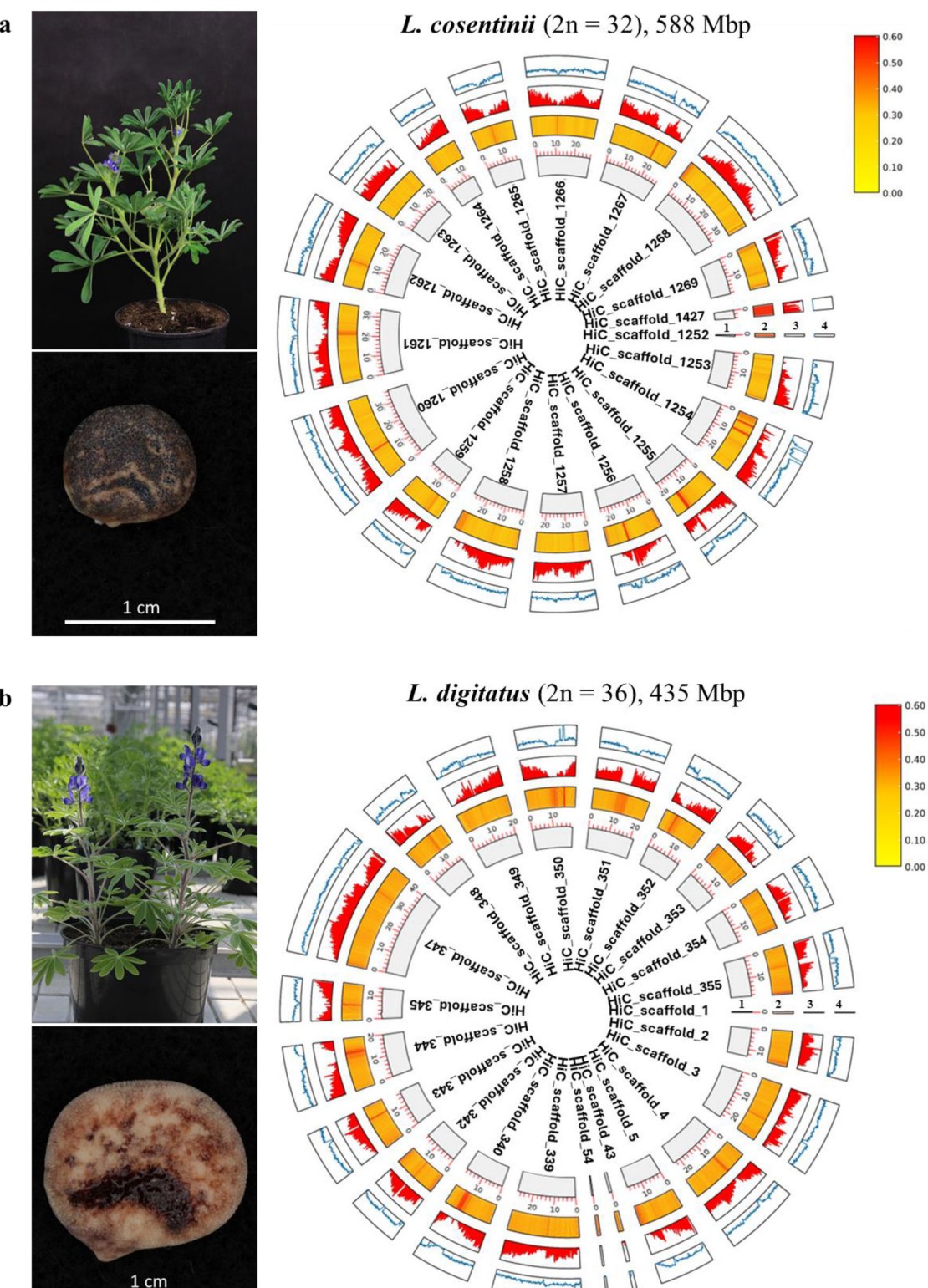

**Fig. 1 | Genome assemblies of two rough-seeded lupins. a** Genome assemblies and annotations of *L. cosentinii*, left panel shows *L. cosentinii* plant and seed. **b** Genome assemblies and annotations of *L. digitatus*, left panel shows *L. digitatus* plant and seed. (1) Circular maps. (2) GC content of the genomes. (3) Gene density. (4) Repetitive element density. For ease of comparison, only scaffolds from *L. cosentinii* and *L. digitatus* are represented here. Source data are provided as a Source Data file.

**Table. 2 | Characteristics of the *L. digitatus*, *L. cosentinii*, *L. albus* and *L. angustifolius* genomes and annotated genes**

| | *L. cosentinii* | *L. digitatus* | *L. albus* | *L. angustifolius* |
|---|---|---|---|---|
| Repetitive content | 352.5 Mbp (60%) | 206.3 Mbp (47.4%) | 255.2 Mbp (56.6%) | 361.0 Mbp (55.3%) |
| No. of predicted genes | 34,780 | 31,260 | 38,258 | 38,545 |
| BUSCO completeness | C: 94.4% [S: 71.4%, D: 23.0%] | C: 93.4% [S: 71.1%, D: 22.3%] | C: 93.3% [S: 72.0%, D: 21.3%] | C: 93.9% [S: 71.5%, D: 22.4%] |
| No. proteins with functional annotation (% of total predicted) | 26,860 (77.2%) | 25,478 (81.5%) | 38,258 (100%) | – |
| No. proteins with GO description (% of total predicted) | 23,544 (67.7%) | 23,019 (73.6%) | 23,423 (61.2%) | – |
| No. proteins with GO description or a functional annotation (% of total predicted) | 30,729 (88.4%) | 28,298 (91.8%) | 38,258 (100%) | 29,225 (75.8%) |

BUSCO statistics: C complete genes, S single-copy, D duplicates.

**Table. 3 | Characteristics of the *L. digitatus*, *L. cosentinii*, *L. albus* and *L. angustifolius* genomes: repetitive DNA**

| | *L. cosentinii* | | *L. digitatus* | | *L. albus* | | *L. angustifolius* | |
|---|---|---|---|---|---|---|---|---|
| | Length (Mbp) | Ratio (%) | Length (Mbp) | Ratio (%) | Length (Mbp) | Ratio (%) | Length (Mbp) | Ratio (%) |
| LINE | 5.20 | 0.88 | 2.79 | 0.64 | 3.70 | 0.82 | 14.3 | 2.19 |
| LTR | 125.1 | 21.3 | 87.6 | 20.1 | 160.1 | 35.5 | 238.9 | 36.6 |
| DNA transposons | 43.0 | 7.30 | 32.1 | 7.37 | 40.8 | 9.05 | 77.5 | 11.9 |
| Simple repeats | 131.9 | 22.4 | 67.4 | 15.5 | 5.01 | 1.11 | 14.0 | 2.14 |
| Others | 17.9 | 3.03 | 13.1 | 3.1 | 17.8 | 3.94 | 25.3 | 3.88 |
| Unclassified | 29.5 | 5.01 | 3.23 | 0.74 | 6.59 | 1.46 | 6.40 | 0.98 |
| Total | 352.5 | 59.9 | 206.3 | 47.4 | 234.0 | 51.9 | 376.3 | 57.6 |

and 93, respectively), whereas *L. digitatus* was characterized by the presence of more insertions (717 in LTR elements and 633 in DNA transposons). *L. cosentinii* featured more deletions in unclassified repeats (1283).

The repetitive DNA graphs for the four lupins showed that DNA transposons were concordant in all four species, especially *L. cosentinii* and *L. digitatus* (Fig. 3), suggesting low divergence between these two genomes. *L. albus* and *L. angustifolius* DNA transposons showed peaks of divergence at ~10% but nevertheless followed the curves of the other two genomes. Conversely, the LTR elements were concordant between *L. albus* and *L. angustifolius*, both of which shared an initial peak at twice the percentage of the other two lupins until 20% divergence. Notably, although *L. cosentinii* and *L. digitatus* showed mostly coherent curves, *L. cosentinii* displayed a peak at ~22% divergence representing 2% of the genome. The LTR elements included in this peak contained 441 deletions (2% of the total deletions).

DNA transposons showed the highest divergence peak (<10%) in *L. albus*, revealing the accumulation and homogenization of new DNA transposons, and that contributions to the total abundance of these elements in the *L. albus* genome are from recently evolved copies. In contrast, the most abundant peaks were observed at 20% for DNA transposons in the genomes of *L. digitatus* and *L. cosentinii*, suggesting that older copies are more abundant than newly evolved copies in these species. However, a discordant repeat landscape was observed in *L. angustifolius*, with two or more ancient peaks for DNA transposons, hinting at an abrupt change and distinct patterns of recently evolved copies of repeat elements. The divergence peak for LTR elements was <10%, suggesting active dissemination and homogenization of these new copies in the genomes of *L. digitatus*, *L. angustifolius* and *L. albus*. In *L. cosentinii*, the peak was observed at 22% divergence, indicating that older copies are more abundant in this genome.

### Consequences of polyploidy during lupin evolution

We anticipated that *L. cosentinii* and *L. digitatus* would show some degree of polyploidy, like other legumes[63]. Accordingly, Genomescope/

Smudgeplot analyses indicated that both species are tetraploid (Fig. 4b, d). A high degree of homozygosity (~99.94%) was evident in both species, shown by the single major peak in the *k*-mer distribution (Fig. 4a, c[64];). The distribution of biallelic single nucleotide polymorphisms (SNPs) in the genome assemblies of both species also indicated tetraploidy. The delta log–likelihood scores, calculated from the difference between the free model and the diploid, triploid and tetraploid models, were 1,202,737, 896,833 and 316,770, respectively in *L. cosentinii*, but 746,976, 428,940 and 101,168, respectively in *L. digitatus* (Fig. 5a, b). The low scores in the tetraploid model therefore favor tetraploidy. The same analysis was applied to individual sequences from both species to determine whether some sequences or chromosomes have a different ploidy to the rest of the genome (a sign of aneuploidy). However, the lowest scores for all sequences again favored the tetraploid model (Fig. 5c, d). Our data therefore indicate that *L. cosentinii* and *L. digitatus* are tetraploid species.

Given the evidence for WGT events (resulting in 2n = 6x = 54), we presumed that lupins with chromosome numbers of 2n = 32 (*L. cosentinii*) and 2n = 36 (*L. digitatus*) possesses basic chromosome numbers of x = 9 if they were subject to multiple chromosome rearrangements, leading changes of 22 and 18 chromosomes (may indicate an entire set of subgenomic chromosomes) in *L. cosentinii* and *L. digitatus*, respectively, due to rediploidization after WGT to establish tetraploidy. We hypothesized that both ECH and LCT might also affect lupin genome evolution. Assuming that the eudicot common ancestor is hexaploid with a basic chromosome number $x = 7$, the legume common ancestor is tetraploid. But ignoring the basic chromosome number, which is proposed to be $x = 11$[44], $x = 16$[23] or various[43]– and a diploid lupin ancestor ($x = 9$)–we propose the scheme shown in Supplementary Fig. 4. In addition, given that the genistoid clade and lupin diploid ancestor had $x = 9$ chromosomes[24,43], we deduced that *L. digitatus* (2n = 36) possesses two diploid subgenomes, indicating a basic chromosome number of x = 9, whereas *L. cosentinii* (2n = 32) might have a basic chromosome number of $x = 8$. However, the basic chromosome number might be $x = 9$ for both species if a progressive

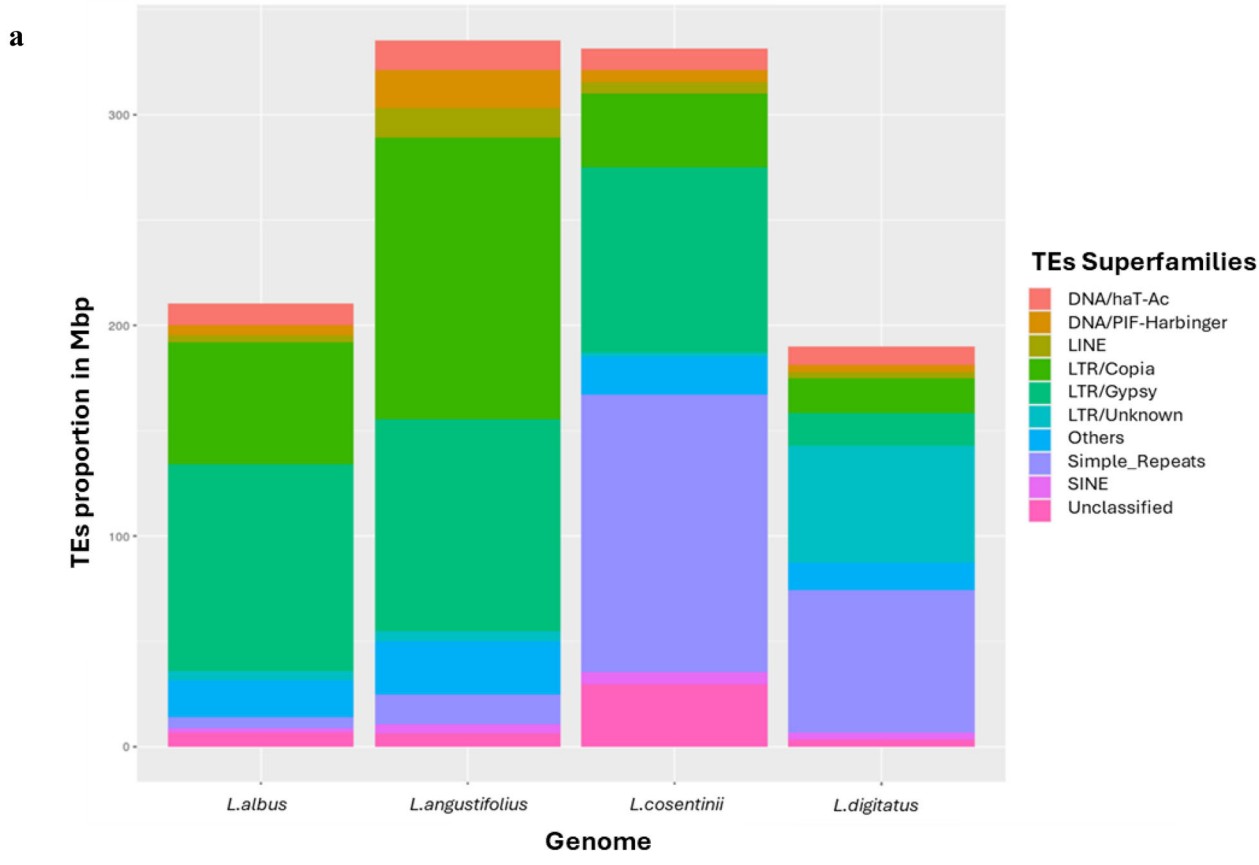

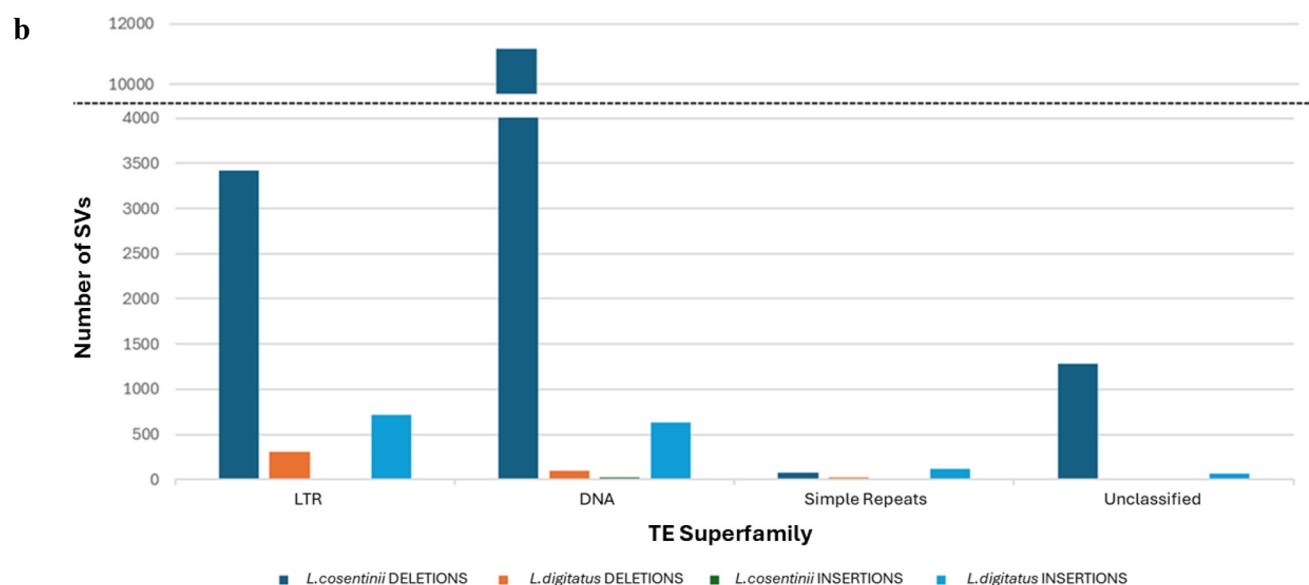

**Fig. 2 | Characteristics of repetitive sequences in lupins. a** Sequence characteristics of repetitive DNA content in the genomes of *L. albus*, *L. angustifolius*, *L. cosentinii* and *L. digitatus*. **b** The distribution of indels (insertions and deletions) in the two assembled genomes, *L. cosentinii* and *L. digitatus*. The distribution was considered in the four main families of transposable elements (LTRs, DNA transposons, simple repeats and unclassified repeats). Source data are provided as a Source Data file.

rediploidization has shaped the current genomes of *L. cosentinii* and *L. digitatus*. We hypothesize that rediploidization may have occurred during the period of evolution separating *L. cosentinii* and *L. digitatus* from *L. albus*, involving changes affecting 18 or 14 chromosomes, respectively. This is supported by the fact that *L. cosentinii* and *L. digitatus* have the smallest chromosome number in the *Lupinus* genus,

indicating the pressure on plants to reduce their chromosome number.

Although we have identified two diploid subgenomes, it was not possible to verify whether they have been shaped by autotetraploidization or allotetraploidization events. This includes numerous rearrangements and rediploidization events that might

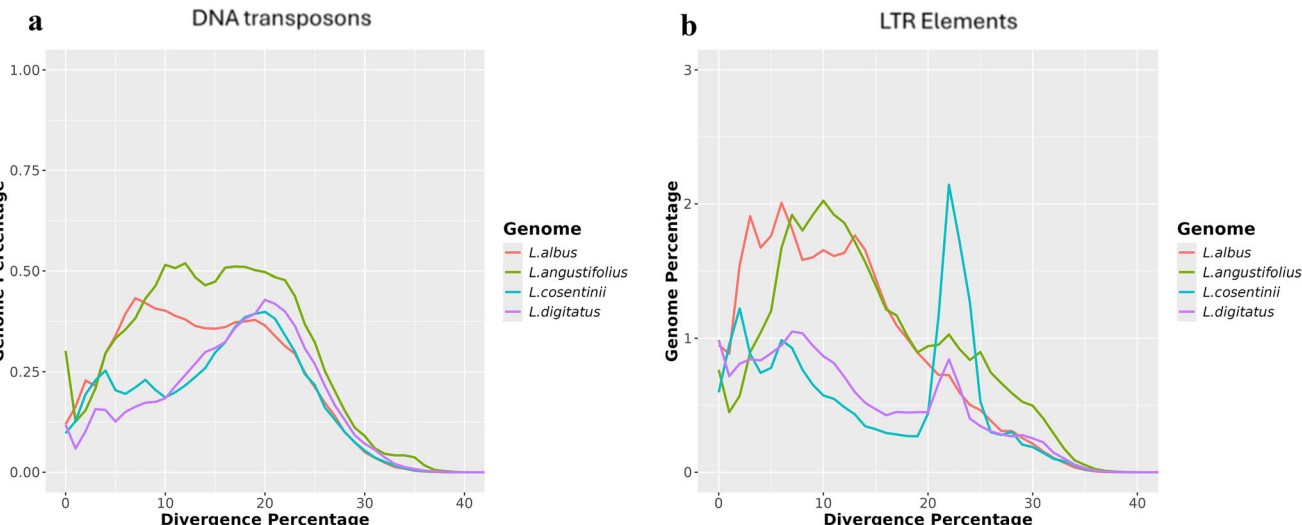

**Fig. 3 | The evolutionary landscape of transposable elements in the four lupins.** **a** DNA transposons. **b** LTR elements. The graphs display the proportion of the genome (%) on the *y*-axis and the degree of divergence based on the kimura distance on the x-axis. The *K* values range from 1 to 40, indicating the level of evolutionary divergence from younger to older transposable elements. Source data are provided as a Source Data file.

span tens of millions of years, exemplifying Old World lupin genome evolution, assuming a phylogenetic relationship in which *L. cosentinii, L. digitatus* and *L. albus* split ~4.5, ~0.5, and ~7.5 Ma, respectively[65]. Rediploidization has been reported in *L. albus*[24], but more studies are needed to demonstrate a polyploidization–rediploidization model in lupins, as previously shown in mangrove species[66].

As two rough-seeded species, *L. cosentini* and *L. digitatus* occupy a position in the *Lupinus* phylogenetic tree (Ainouche and Bayer 1999) that separates smooth-seeded lupins into two groups, suggesting they are genetically closer to *L. albus* (one of the closest smooth-seeded lupins to rough-seeded species) than *L. angustifolius*[65] but also indicating unique evolutionary changes. Indeed, these species adapted to semi-desert and warm Mediterranean conditions, and facilitated their survival in rapidly changing environments by increasing genomic plasticity. Polyploidization may have promoted the diversification of lupins, as shown by the unique morphology of rough-seeded lupins, with scabrous-tuberculate testa[48]. The seed alkaloid content of rough-seeded lupins is moderate while growing naturally or though long selection, giving hope the untreated seeds of rough-seeded lupins could be used directly as food/feed[4,67].

Polyploidization happened early in the evolution of the genistoid clade, indicating that WGD may have predated the divergence of Old World and New World lupins[43]. The chromosome number of rough-seeded species is small, like the species found in South America. Additionally, the annual American lupins are the only species with chromosome numbers of 2n = 32 or 34[34]. This might suggest the chromosome number 2n = 36 arose independently at least twice within the genus, an indication of convergent evolution in the context of ploidy. On the other hand, the chromosome number in Old and New World lupins may raise questions about the geographical origin of lupins, which is proposed to be in the Old World[68]. In the northern hemisphere, where the genistoid clade is thought to have originated during the Paleocene epoch[38], evolutionary studies of these two groups of lupins will shed new light on the evolution of the entire *Lupinus* genus. The rough-seeded lupins described here can be used as a model to investigate the evolution of American lupins.

The complex evolution of the *Lupinus* genus is characterized by remarkable diversity in genome size, basic and somatic chromosome numbers, and chromosome rearrangements, in contrast to other legume genera. For example, *Phaseolus* and *Cajanus* (phaseoloids) feature mostly diploid species with the same chromosome number 2n = 22, whereas *Arachis* (dalbergioids) features both diploids (2n = 20) and tetraploids (2n = 40), and *Dalbergia* (dalbergioids) features exclusively diploid species with the chromosome number 2n = 20[69]. Interestingly, the *Lupinus* diploid ancestor with a basic chromosome number of $x = 9$[24] was confirmed across the genistoids[38]. However, the basic chromosome number may differ in early-diverging genistoid species, including those in the genus *Sophora* such as *S. flavescens* (2n = 18, diploid)[70] and *S. japonica* (2n = 28, ploidy unreported)[71], and also *Crotalaria* spp. (2n = 14, 16, or 32)[72]. Furthermore, the genistoid genus *Ulex* has a chromosome number of 2n = 32, 64 or 96 and *Genista* has a chromosome number of 2n = 48, 44 (described as aneuploid), 72 or 96[73]. *L. digitatus* (2n = 36) is the only known Old World lupin providing a direct example of $x = 9$, corresponding to the *Lupinus* diploid ancestor. In contrast, *L. cosentinii* has a different basic chromosome number ($x = 8$ or 9) and refutes the hypothesis that species with chromosome numbers such as 2n = 32 are aneuploids[36,68] or underwent various chromosomal rearrangements. However, $x = 8$ is considered a primitive basic number of the genistoids.

## Comparative genomics in lupin species

We compared our *L. cosentinii* and *L. digitatus* genome assemblies to the annotated and curated *L. albus*[23] and *L. angustifolius*[22] genome assemblies. Systematic pairwise comparisons revealed large syntenic blocks conserved in all four genomes (Fig. 6, Supplementary Figs. 5–10). The largest blocks were 24.3, 19.9, 17.3, 10.9, 9.9 and 9.8 Mbp in length, consisting of 1649, 783, 1190, 775, 593 and 508 collinear genes in the *L. digitatus* vs *L. cosentinii* (LdLc), *L. digitatus* vs *L. albus* (LdLa), *L. albus* vs *L. cosentinii* (LaLc), *L. angustifolius* vs *L. cosentinii* (LnLc), *L. angustifolius* vs *L. digitatus* (LnLd) and *L. albus* vs *L. angustifolius* (LaLn) comparisons, respectively (Supplementary Fig. 11). The degree of duplication showed a similar distribution when considering the total number of genes and genes located in smaller syntenic blocks. The average degree of duplication when considering all genes was similar in *L. cosentinii* (1.36) and *L. digitatus* (1.35) but increased to 1.43 in both species when considering the four smaller syntenic blocks (Supplementary Table 1). The rate of synonymous substitutions ($K_s$) calculated for duplicated BUSCO genes suggested that *L. cosentinii* and *L. digitatus* are more closely related to each other than the other genome combinations, as confirmed by the LdLc density curve (light blue) being lower than the others. In contrast, the LaLn (green) and LnLc (blue) density curves were the highest peaks in the graph,

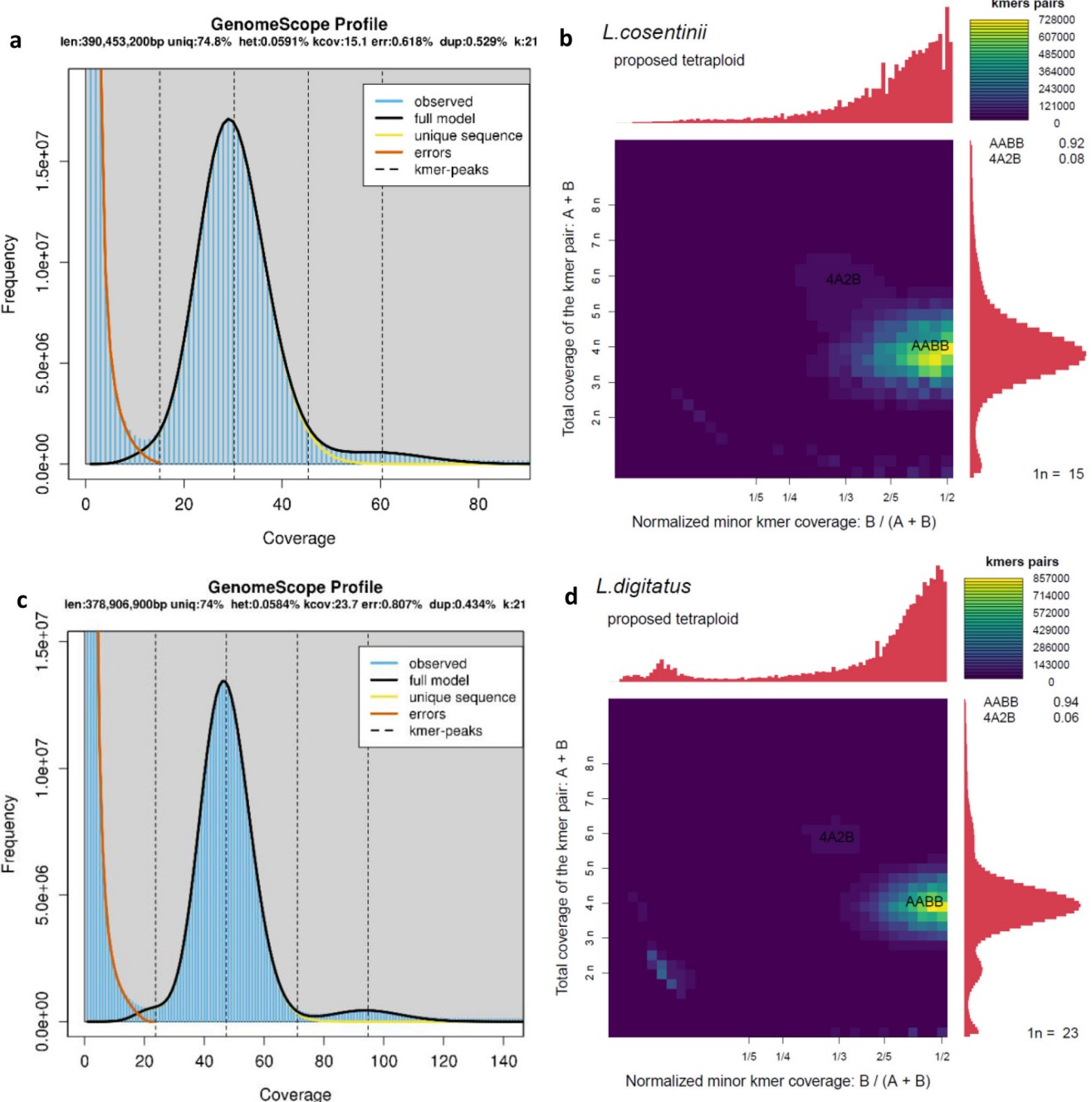

**Fig. 4 | The k-mer distributions and proposed ploidy levels of *L. cosentinii* and *L. digitatus*. a** *L. cosentinii* GenomeScope profile. **b** *L. cosentinii* k-mer distribution. **c** *L. digitatus* GenomeScope profile. **d** *L. digitatus* k-mer distribution.

suggesting that the relationship between *L. albus* and *L. angustifolius* and that between *L. angustifolius* and *L. cosentinii* are the most distant among the pairwise combinations (Fig. 7a).

To explore the evolution of the gene families among the four species, we used orthologous clustering to define 25,663 gene families (Fig. 7b). Most (19,203) were common to all the species, followed by the gene families shared by *L. albus*, *L. cosentinii* and *L. digitatus* (3221). The number of single-species gene families was similar in *L. cosentinii* and *L. digitatus* (94 and 88, respectively). The species tree inferred by Orthofinder indicated that the most closely related species were *L. digitatus* and *L. cosentinii*, followed by *L. albus* (Fig. 7c), reflecting the relationship known so far about *Lupinus* species (e.g. Drummond et al.[65]). The number of expanded gene families (2784 and 2751 in *L. cosentinii* and *L. digitatus*, respectively) and the number of contracted

gene families (4071 and 4075, respectively) were similar in the two new assemblies when compared to *L. albus*. The same numeric similarity was observed when we used *L. angustifolius* as the reference species (Supplementary Table 2).

In conclusion, we have described whole-genome assemblies of the rough-seeded lupin species *L. cosentinii* and *L. digitatus*, providing insight into lupin genomics and evolution, and adding to the genetic resources available for lupin breeding and crop improvement. These two annotated assemblies provide key reference genomes for lupin and, more generally, the genistoid clade. Importantly, we provide evidence that both species are tetraploid. Our data provide insight into the role of genome duplication during lupin evolution but further evidence from other wild and domesticated species would help to complete the picture, enabling us to understand the domestication,

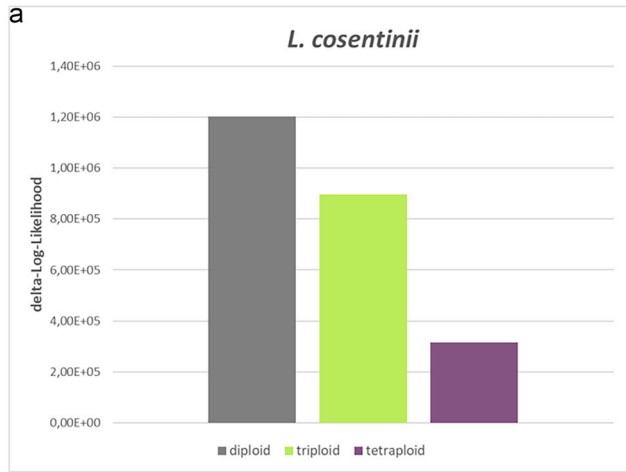

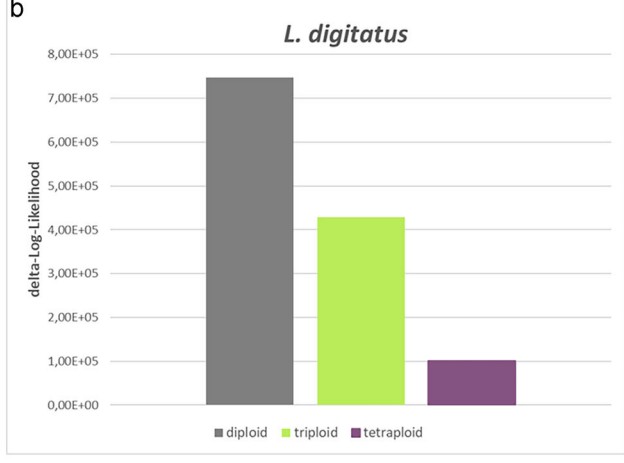

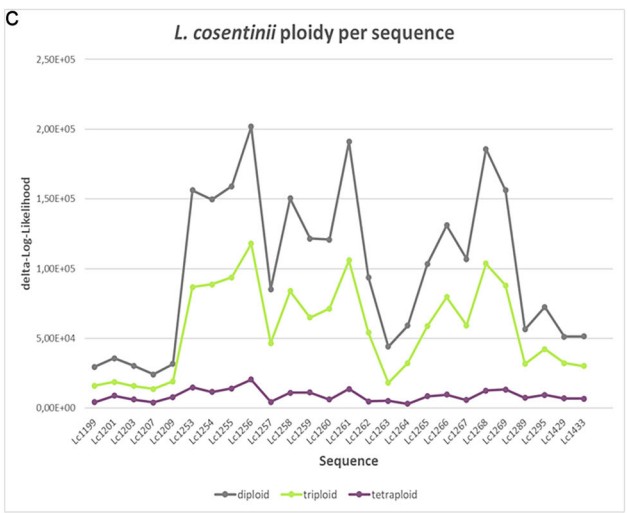

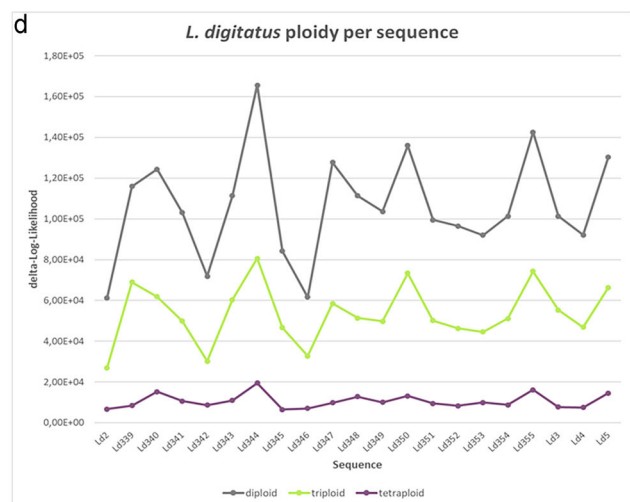

**Fig. 5 | Ploidy prediction in *L. cosentinii* and *L. digitatus* according to nQuire.** The lower the delta log–likelihood score, the better the fit to the corresponding model. **a** Prediction scores on the *L. cosentinii* whole genome assembly and **b** on the longest 21 sequences (N90). **c** Prediction scores on the *L. digitatus* whole genome assembly and **d** on the longest 26 sequences (N80). Source data are provided as a Source Data file.

agricultural improvement, environmental adaptability and evolution of legume crops, and facilitating the exploitation of legumes as part of a healthy and sustainable diet. The analysis of lupin gene families provided insights into their relationship with phenotypic diversification and species adaptation, which will facilitate the exploitation of underutilized legume species by identifying genes that can be used in crop breeding programs. Our work will underpin the development of improved lupin crops by exploiting the genetic diversity of CWRs and orphan crops to promote the conservation and sustainable utilization of lupins as a source of high-quality dietary protein, and to promote the domestication of a greater variety of wild lupin species.

## Methods
### Plant materials
The characteristics of *L. cosentinii* and *L. digitatus* are summarized in Supplementary Table 3. We selected *L. cosentinii* 98460 based on its seed production to secure enough seeds for further research and multiplication. Accession *L. cosentinii* 98460, CV population, country of collection Morocco, was obtained from the Polish *Lupinus* Collection (Poznan Plant Breeders Ltd, Wiatrowo branch, Poland). We used the only *L. digitatus* accession provided by US Department of Agriculture (ID: PI 660697, collected in Spain). For both species we developed single-seed descent (SSD) lines, and then multiplied them to conserve genetic resources. Seeds of both species were scarified,

vernalized for 21 days and then sown in 7.5-L pots containing a 1:1 mix of peat and vermiculite. The plants were grown in a phytotron at 22/18 °C (day/night temperature) with a 16-h photoperiod, 60–65% relative humidity, and watering as required.

### Extraction of high-molecular-weight DNA
For PacBio sequencing, high-molecular-weight DNA was extracted from 1 g frozen young leaf material that was ground to powder under liquid nitrogen. Nuclei were isolated in NIBTM buffer (10 mM Tris, 10 mM EDTA, 0.5 M sucrose, 80 mM KCl, 8% PVP (MW 10 kDa), 100 mM spermine, 100 mM spermidine, pH 9.0) supplemented with 0.5% Triton X-100 and 0.2% 2-mercaptoethanol, followed by filtration through 100-µm and 40-µm cell strainers and centrifugation (2500 g, 10 min, 4 °C)[74]. DNA was then extracted from nuclei using the Genomic-tip 100/G kit (Qiagen) and eluted in low-EDTA TE buffer (10 mM Tris, 0.1 mM EDTA, pH 9.0). DNA size and integrity were analyzed by pulsed-field gel electrophoresis (PFGE) using the CHEF Mapper system (Bio-Rad Laboratories) with a 5–450 kbp run program. DNA was quantified using the Qubit DNA BR Assay Kit and a Qubit fluorimeter (Thermo Fisher Scientific) and its purity was evaluated by spectrophotometry using a Nanodrop 2000 (Thermo Fisher Scientific). PacBio libraries were prepared from both species using the SMRTbell prep kit v3.0, followed by SMRT sequencing on a Sequel II device (Pacific Biosciences).

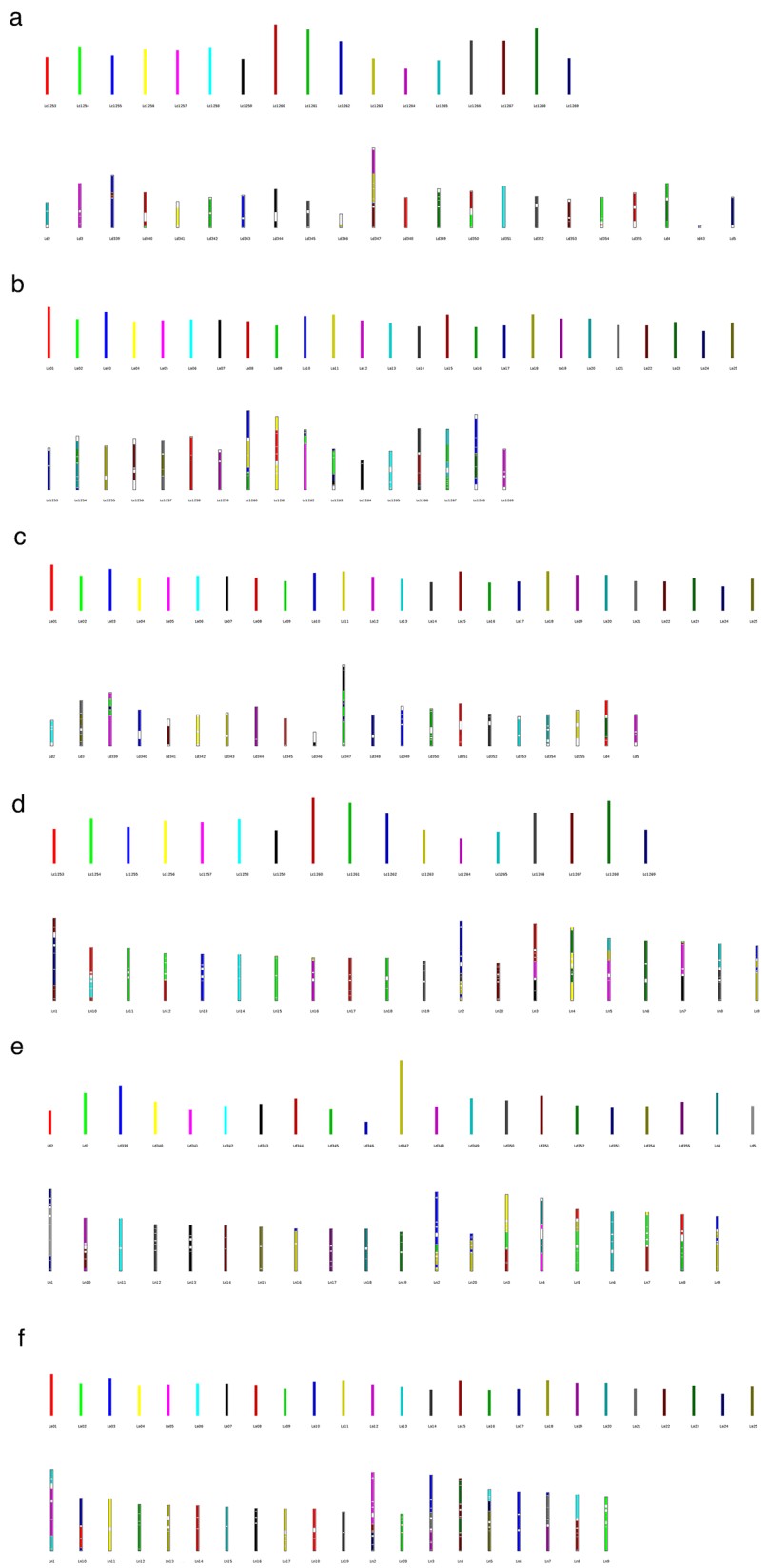

**Fig. 6 | Collinearity analysis. a** Collinear blocks between *L. digitatus* (Ld) and *L. cosentinii* (Lc). **b** Collinear blocks between *L. albus* (La) and *L. cosentinii* (Lc). **c** Collinear blocks between *L. albus* (La) and *L. digitatus* (Ld). **d** Collinear blocks between *L. angustifolius* (Ln) and *L. cosentinii* (Lc). **e** Collinear blocks between *L. angustifolius* (Ln) and *L. digitatus* (Ld). **f** Collinear blocks between *L. angustifolius* (Ln) and *L.albus* (La). Source data are provided as a Source Data file.

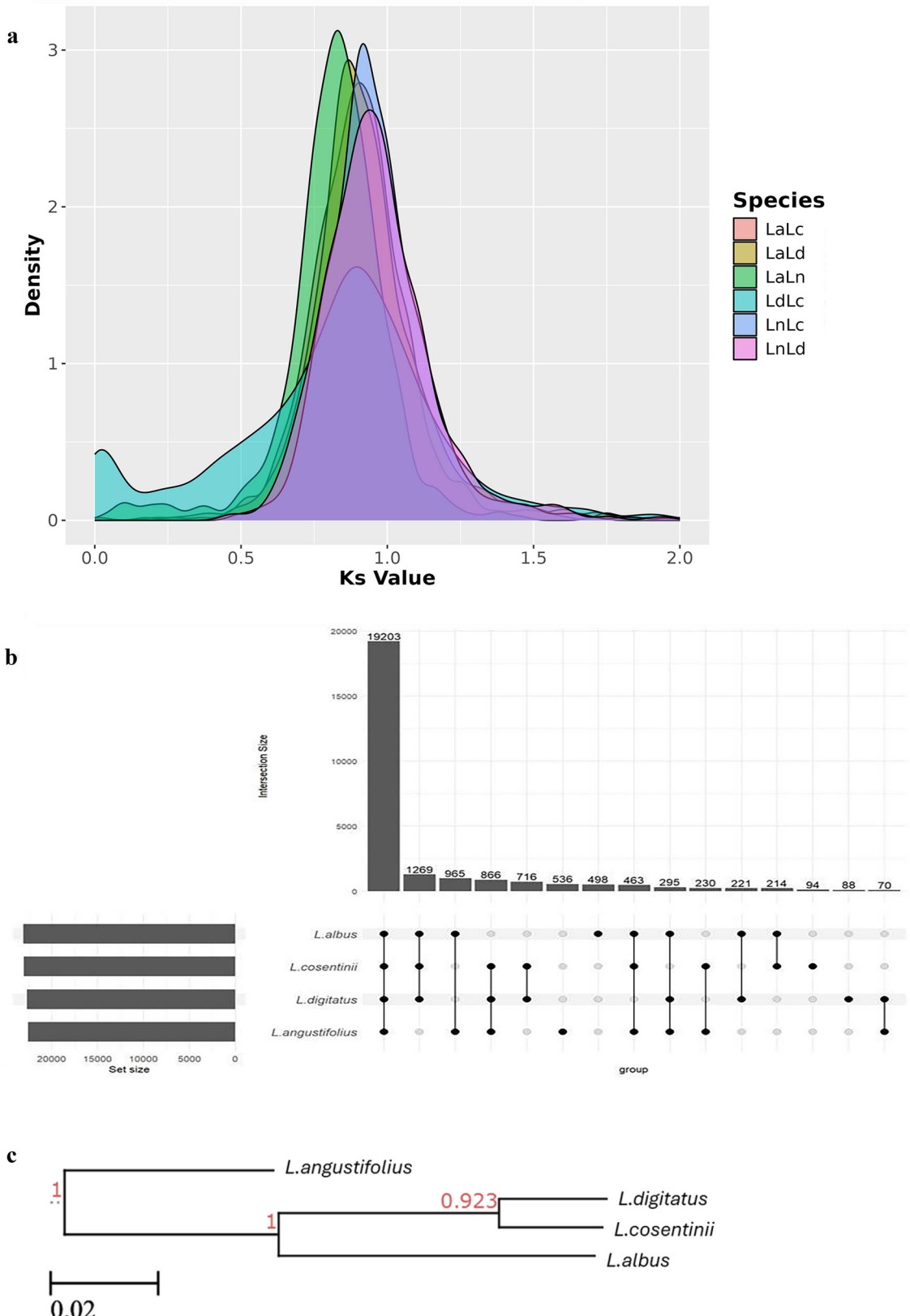

**Fig. 7 | Comparative analysis of synteny and gene families. a** Distribution of $K_s$ scores calculated from the coding sequences of orthologous genes from the six comparisons. The $K_s$ values are shown on the x-axis, and their density is on the y-axis. **b** UpSet plot comparing the shared gene families (orthogroups) among the four lupin species. The histogram displays the number of gene families shared by the genomes indicated below the x-axis. The histogram is sorted by the number of shared families in each possible combination of the four genomes. **c** A phylogenetic tree constructed for the four lupin species using the STAG method in OrthoFinder. The values indicate the branch lengths. The scale bar represents the number of differences between the sequences. Source data are provided as a Source Data file.

## Whole-genome library preparation for Illumina sequencing

We fragmented 700 ng of high-molecular-weight DNA using an S220 sonicator (Covaris) and a WGS library was generated for both species using the KAPA Hyper Prep kit with a PCR-free protocol according to the manufacturer's instructions (Roche). We applied final size selection by using a 0.7-fold ratio of AMPureXP beads (Beckman Coulter). The sequence length was assessed by capillary electrophoresis on a 4150 TapeStation (Agilent Technologies) and the library was quantified by qPCR against a standard curve with the KAPA Library Quantification Kit (Roche). Libraries were sequenced on a Nova-Seq6000 Illumina platform in 150PE mode.

## DNA extraction and Bionano optical mapping

Ultra-high-molecular-weight DNA was extracted from fresh sprouts or leaves (<2 cm in length) of *L. cosentinii* 98460 and *L. digitatus* PI 660697, which were kept in the dark for ~16 h before extraction[75]. DNA was isolated from ~0.4 g of sprouts using the Bionano Prep High Polysaccharides Plant Tissue DNA Isolation Protocol (Bionano Genomics, document number 30128, revision C). For each species, two agarose plugs were prepared according to Staňková et al.[76]. DNA extracted from one plug was assessed for length and concentration by PFGE as above. DNA from the second plug was used for Bionano optical mapping following the direct label and stain (DLS) protocol (Bionano Genomics). The labeled and stained DNA was loaded onto a Bionano Saphyr (Bionano Genomics).

## Hi-C library preparation for Illumina sequencing

Hi-C libraries were prepared from 0.52 g of frozen young leaves of *L. cosentinii* 98460 and *L. digitatus* PI 660697 using the Proximo Hi-C (Plant) Kit and protocol v4.0 (Phase Genomics), incorporating three additional wash steps and 12 PCR amplification cycles. The quality of the Hi-C libraries was assessed using a D1000 ScreenTape Kit on a 2200 TapeStation (Agilent Technologies), and the quantity was determined by qPCR using primers that anneal to the adapter sequences. Libraries were sequenced on a NovaSeq 6000 Illumina platform in 150PE mode.

## RNA-Seq library preparation for Illumina sequencing

We prepared 22 RNA samples from *L. cosentinii* 98460 (six samples of young leaves, four of fully developed leaves, four petioles, four pods and four roots) as well as 32 from *L. digitatus* PI 660697 (six of leaves, six petioles, six pods, five stems, four apical stems, three lateral roots and two main roots). Total RNA was isolated from 30 mg of ground plant tissue using the SV Total RNA Isolation System Kit (Promega) and its concentration and integrity were assessed using the RNA 6000 Nano Kit on a Bioanalyzer (Agilent Technologies). All samples showed an RNA integrity number (RIN) > 7. Samples were quantified using the Qubit RNA HS Assay Kit (Thermo Fisher Scientific). We pooled 2–3 RNA samples from the same tissue for library preparation to make pools of five different *L. cosentinii* tissues (young leaves, fully developed leaves, petioles, pods and roots) and seven different *L. digitatus* tissues (leaves, pods, stems, lateral roots, main roots, petioles and apical stems). RNA-Seq libraries were generated using the TruSeq stranded mRNA ligation kit (Illumina) from 1000-ng RNA samples, after poly(A) capture and according to the manufacturer's instructions. Library quality and size were assessed by capillary electrophoresis using a 4150 TapeStation as above, and their quantity was determined by real-time PCR against a standard curve using the KAPA Library Quantification Kit as above. The libraries were pooled at equimolar concentrations and sequenced on a Novaseq6000 device in 150PE mode.

## De novo genome assembly from PacBio Hi-Fi reads

PacBio Hi-Fi reads were assembled de novo using HiCanu v2.1.1[77] with default parameters. Completeness was evaluated using BUSCO v5.4.7[78] and the Fabales_obd10 database comprising 5,366 genes. Illumina

WGS data were evaluated using FastqQC v0.11.9 and low-quality segments and sequencing adapters were removed using Fastp v0.21.0[79]. Filtered reads were aligned on the assembly using bwa-mem2 v0.7.17 and residual base-level errors were corrected by three rounds of polishing using Pilon v1.23. To evaluate the effectiveness of this approach, we applied variant calling using GATK HaplotypeCaller v4.2.2[80] before and after polishing. We also used purge_haplotigs v1.1.2[81] to remove putative haplotype duplications. BLAST v2.9.0+[82] was used to screen all remaining reads against the NCBI nr database to confirm that all reads belonged to the kingdom Viridiplantae, thus ensuring the absence of contamination. BLAST results were filtered considering a minimum identity coverage of 80% and minimum query coverage of 40%. BLAST was also used to screen mitochondrial (https://ftp.ncbi.nlm.nih.gov/refseq/release/mitochondrion/) and chloroplast (https://ftp.ncbi.nlm.nih.gov/refseq/release/plastid/) RefSeq databases and published *L. albus* organelle sequences[23] to exclude organelle DNA.

## Scaffolding with Bionano optical maps

Bionano sequencing outputs were filtered to remove molecules <150 kbp in length before de novo assembly and alignment on the corresponding genome maps using Bionano Solve v3.7.1 (https://bionanogenomics.com/support/).

## Chromosome-level scaffolding with Hi-C data

The Hi-C raw reads were aligned on the Bionano genome assemblies using the Juicer v1.6 pipeline[83] before a second round of scaffolding using 3d-dna v18.09.22[84] with default parameters. Before the misjoin correction step, the Hi-C contact matrix was manually curated with juicebox v1.11.08.

## Structural annotation

Repetitive elements in all four genomes were identified with HiTE v3.2[85]. To evaluate the presence of indels in the two assembled genomes, Illumina reads from the polishing step were aligned and structural variations were called in the repeated regions of the genome by applying the TEPID pipeline v0.15[86,87]. Genomic divergence was computed with Parsing-RepeatMasker-Outputs v5.8.2[88].

RNA-Seq data were aligned on the assembled genome using Hisat2 v2.2.1[89] with a maximum intron length of 60 kbp. The alignments were then converted into intronic hints, retaining only those supported by at least 10 reads. RNA-Seq data were also assembled into transcripts using Trinity v2.15[90]. Only the primary isoform of all reconstructed genes, namely those classified as 'main' and 'complete' by Evidential Gene v2018, were retrieved and aligned on the assembled genome using gmap v2017-11-15[91] for use as extrinsic evidence. Finally, proteins from the closely related species *L. albus* (https://phytozome-next.jgi.doe.gov/info/Lalbus_v1) were aligned on the genome assembly using Genome Threader v1.7.1[92]. The extrinsic evidence extracted from the three different sources described above was then used for final ab initio gene prediction with Augustus v3.3.3[93] trained using *Fabales* BUSCO genes (BUSCO v5.4.7, Fabales_odb10 database). The predicted genes were filtered using InterProScan v5.52-86.0[94] to identify genes structurally related to known protein domains.

## Functional annotation

Genes were functionally annotated based on the analysis of homology (BLAST v2.9.0+, keeping only the best hits for each gene) and protein domains (InterProScan). For homology-based analysis, we considered three levels of confidence: (1) genes with functional annotations in SwissProt (https://www.uniprot.org/uniprotkb?facets=reviewed%3Atrue&query=%2A), RefSeq plant databases (https://ftp.ncbi.nlm.nih.gov/refseq/release/plant/) and/or TAIR were labeled as high confidence; (2) genes were labeled as medium confidence if we retrieved functional annotations based on the *L. albus* proteome; and (3) genes that were not annotated using the first two levels were screened

against the NCBI nr database to obtain a descriptive annotation. The alignments were filtered by percentage coverage and identity, both with thresholds of 80%. GO terms were derived from homology-based analysis at the first and second confidence levels (if the function was concordant) and from InterProScan analysis.

## Ploidy analysis

The level of ploidy in *L. cosentinii* and *L. digitatus* was assessed using two methods, the first based on *k*-mer distribution and the second on biallelic SNP frequencies, applied to Illumina reads after noise reduction. For the first approach, the *k*-mers in WGS Illumina reads were counted using KMC v3.2.2[95]. The *k*-mer distributions were analyzed using Genomescope2.0 and Smudgeplot[96] with parameter –homozygous due to the high level of homozygosity. For the second approach, Gaussian mixture models were used to estimate the ploidy level with nQuire[97]. Reads were mapped to the genome and biallelic SNP frequencies were calculated. A delta log-likelihood score was then calculated between a free model and three fixed models (diploid, triploid and tetraploid). The lowest fixed-model score points to the most likely ploidy. This analysis was applied to the whole *L. cosentinii* and *L. digitatus* datasets and also to 26 individual sequences in *L. cosentinii* (corresponding to ~80% of the genome assembly) and 21 in *L. digitatus* (corresponding to ~90% of the genome assembly). If some sequences showed a lower score in a different ploidy model than the rest of the genome, this could be interpreted as a sign of aneuploidy. The rationale behind the use of two approaches was to validate the predicted ploidy level independently from the homozygosity of the two assembled genomes.

## Comparative genomics

Orthofinder v2.5.4[98] was applied to all four species with default parameters (-S diamond). Genes in an orthogroup from the same species were considered paralogs and members of the same gene family. The file N0.tsv inside the "Phylogenetic Hierarchical Orthogroups" folder was used for downstream analysis, representing the different gene families. A phylogenetic tree was built based on the OrthoFinder results and converted to its ultrametric format using treePL.

Synteny was evaluated using MCScanX[99] with default parameters. Specifically we used MCScanX_h, allowing the exploitation of orthologous genes from *L. albus*, *L. angustifolius*, *L. cosentinii* and *L. digitatus* predicted by Orthofinder. We tested the pairwise comparisons *L. digitatus* vs *L. cosentinii* (LdLc), *L. albus* vs *L. digitatus* (LaLd), *L. albus* vs *L. cosentinii* (LaLc), *L. abus* vs *L. angustifolius* (LaLn), *L. angustifolius* vs *L. cosentinii* (LnLc) and *L. angustifolius* vs *L. digitatus* (LnLd). The $K_s$ distribution was evaluated considering only duplicated BUSCO genes. The coding regions of the orthologous gene pairs from the six pairwise comparisons were used to calculate $K_a/K_s$ ratios in the MCScanX downstream analysis package "add_kaks_to_synteny".

Variation in gene family sizes were characterized using Cafe5 v5.1.0[100] with the -k 7 parameter followed by GO functional enrichment analysis of the expansion (gain) and contraction (loss) events in the gene families. Cafe5 was applied to all four species and the evaluation of gene family expansion/contraction and GO enrichment were achieved by comparing each assembled genome against one of the two published genomes, independently. GO enrichment analysis was implemented using the 'enricher' method of the clusterProfiler library[101], considering only significant results ($p < 0.05$).

## Reporting summary

Further information on research design is available in the Nature Portfolio Reporting Summary linked to this article.

## Data availability

The raw sequence read data generated in this study have been deposited in the Sequence Read Archive (SRA) of the National Center of Biotechnology Information (NCBI) under BioProject ID PRJNA1080360 (*L. cosentinii* and *L. digitatus*) [https://www.ncbi.nlm.nih.gov/search/all/?term=PRJNA1080360], and Biosample IDs SAMN40127157 (*L. cosentinii*) [https://www.ncbi.nlm.nih.gov/biosample/?term=SAMN40127157] and SAMN40126867 (*L. digitatus*) [https://www.ncbi.nlm.nih.gov/biosample/?term=SAMN40126867]. The genome assemblies and annotations are publicly available under the BioProject ID PRJNA1080360 and can also be accessed at Figshare: *L. cosentinii* [https://doi.org/10.6084/m9.figshare.25367899] and *L. digitatus* [https://doi.org/10.6084/m9.figshare.25367935]. Seeds of *L. cosentinii* and *L. digitatus* are available upon request. Source data are provided with this paper.

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

## Acknowledgements

This study was supported by the National Science Centre, Poland (grant nos. HARMONIA 7 2015/18/M/NZ2/00422 and OPUS 18 2019/35/B/NZ8/04283 to KS). We thank Andrea Benazzo and Robert Hasterok for critical comments that improved the manuscript. We acknowledge the support provided by the Horizon 2020 Project INCREASE, grant agreement number 862862 (R.P. and K.S.; https://www.pulsesincrease.eu). INCREASE has received funding from the European Union's Horizon 2020 research and innovation programme under grant agreement no. 862862. This publication reflects only the author's view and neither the Research Executive Agency (REA) nor the European Commission are responsible for any use that may be made of the information it contains.

## Author contributions

K.S. conceptualized the study, designed the experiments, wrote the manuscript, interpreted data and supervised the project. L.V. carried out bioinformatic analysis, prepared the figures, and helped to write the manuscript. M.T. cultivated the plants under controlled conditions. M.K. analyzed the plants, extracted nucleic acids, assisted with data interpretation and manuscript preparation. E.F assisted with the bioinformatic analysis and drafted the corresponding part of the manuscript. E.C. and A.R.L. performed laboratory experiments. U.K.T. assisted with bioinformatic analysis and manuscript preparation. H.J. assisted with bioinformatic analysis. M.D. supervised genome sequencing and assembly. M.D., M.N.N., P.B., D.E., R.P. and S.A.J. assisted with data interpretation. R.P., M.D. and S.A.J. contributed to the substantive revision and editing of the manuscript. All authors read and approved the final version of the manuscript.

## Competing interests

The authors declare no competing interests
