## [Peer Review file · Nature Communications]

The unexplored diversity of rough-seeded lupins provides rich genomic resources and insights into lupin evolution

Corresponding Author: Dr Karolina Susek

Version 0:

Reviewer comments:

Reviewer #1

(Remarks to the Author)

Karolina and co-authors have assembled the genomes of two wild lupins. Based on this, they analyzed the polyploidy of the two species, the characteristics of these expanded genes, and the possible association with the large seed size of the two species. I see the limit value of the work as a resource, but the overall analysis/quality of the manuscript is low/rough. I have some comments for the authors to improve the manuscript.

1. In the abstract, the authors say that the two wild lupins have a large seed size, reflecting convergent evolution contributing to lupin domestication. Does the domesticated lupin *L. albus* have a much larger seed size? What is the convergent trait that evolved in which species for its domestication?
2. When presenting the relationships of the various wild lupins, why do the authors divide them into two groups, New World and Old World? This isn't a scientific grouping of biological entities. Is the geographical distribution of lupins related to the activity of Europeans?
3. I see that you talk more about "New World and Old World lupins" in the conclusion section. I suggest that the authors move this part from the conclusion to the discussion section after the polyploidy study.
4. Since there are 16 pairs ($2n=32$) and 18 pairs ($2n=38$) of chromosomes for *L. cosentinii* and *L. digitatus* respectively, why did the authors obtain 19 and 22 large scaffolds after two rounds of scaffolding?
5. I'm a bit surprised that simple repeats are the largest classes (22.5% and 15.5%) of repetitive sequences, what kind of simple repeats were extensively amplified, how about these in other closely related species?
6. The authors assume that the basic chromosome number for *L. cosentinii* and *L. digitatus* is $x=8$ and $x=9$, respectively. What is the method and process of deduction? Are the tetraploidization events ancient or recent, i.e. when did they occur (if the two parental sub-genomes rearranged after tetraploidization and became diploid again)? Is it a common tetraploidization event shared by the two wild lupins before they diverged, or did it occur independently after they diverged? Is it an auto- or allo-tetraploidization event?
7. The authors state that they have performed a genomic synteny analysis between *L. cosentinii*/*L. digitatus* and *L. albus*. As mentioned, *L. albus* experienced a WGT event, I'm concerned about how these 3:2 genomic relationships were compared (3 means three subgenomes generated by WGT in *albus*, 2 means two subgenomes generated by WGD in *cosentinii digitatus*)?
8. For the part of the expanded genes associated with the trait, and the GO analysis. I suggest to largely remove them, as they just do not make sense, providing some real features that can be obtained by genomic comparative analysis is much better.
9. There are two similar citations in the second paragraph of the introduction (Hufnagel et al. 2020)(Hufnagel et al. 2020),

please correct.

10. The last sentence in the last paragraph of the Introduction is very abrupt. I suggest that the authors briefly introduce the basic study and results before this sentence.

11. Ks distributions estimate the time of sequence/species divergence, does this mean that LdLc are much closer evolutionarily?

Reviewer #2

(Remarks to the Author)

The manuscript by Susek and colleagues describes the genome sequencing, assembly and annotation of two Old World Lupin species in *Lupinus cosentinii* and *L. digitatus*, bringing the total number of Old World Lupin genome assemblies to 4 (*L. albus* and *L. angustifolius* are the other two species with chromosome level assemblies). The authors determine that *Lupinus cosentinii* and *L. digitatus* are tetraploid and subsequently conduct some comparisons to the *L. albus* assembly only. The latter seems odd given that there are two quality genome assemblies and annotations available to conduct these comparisons to. I encourage the authors to expand these comparisons to *L. angustifolius*. The authors should also include line numbers to make the review and feedback on the manuscript easier.

Whilst the manuscript is technically sound, more detail in the methodology is required and the quality of the Figures could improve as per my comments below. Furthermore, some results are described in the methodology which should be moved to the results. I congratulate the authors on the generation of two assemblies for these two Old World Lupin species.

Specific comments:

Page 3 Lines 6-8 Define what you mean by wild relatives. This is important as the subsequent sentence suggests you refer to secondary or tertiary gene pools of a genus rather than wild, undomesticated accessions for a species.

Page 3 lines 8-11. There are whole genome sequences available for wild relatives of chickpea please expand this section to ensure it is accurate and cite the wild Cicer publications.

Page 3 lines 12-15. There is also a second *albus* lupin assembly. The Hufnagel assembly is cited twice. Please correct and provide the correct reference for the other assembly.

Page 3 lines 21-25 It would be good to clarify which traits you refer to when it comes to the domestication of a lupin species and references supporting the level of domestication should be provided.

Page 5 de novo assemblies. Why was HiCanu selected as the preferred assembly tool? Were other genome assembly packages suitable for polyploid genomes assessed (e.g. hifiasm, ALLHiC, and PolyGembler)?

Page 5 de novo assemblies. With all the data generated one would have thought to be able to scaffold the assembly to level that matches the number of chromosomes. For both the *L. albus* and *L. angustifolius* genomes using a combination of long and short read data coupled with either BioNano or Hi-C data were able to achieve this. A Hi-C contact map and BioNano maps should be presented. The authors should compare the statistics of the assemblies to the current *L. albus* and *L. angustifolius* genome statistics. The same goes for the gene content and repetitive sequences. Simply only comparing to *albus* lupin is not sufficient.

Page 7 Comparative genomics. Why only compare to *Lupinus albus* and not both sequenced lupin species? As they are all Old World species it seems strange to omit one species in the comparisons.

General comment on the methods section: consistently provide details of manufacturers. The authors currently provide manufacturer details for some but not other reagents/instruments used. For example, no supplier provided for Bionano, TapeStation or AMPureXP beads.

Page 10 "Plant materials" Provide details of the cultivar/accessions selected as the representatives of *L. cosentinii* and *L. digitatus* for which you sequenced the genomes and how /from where were these obtained?

Page 11 Lines 18-19. Delete "generating ~32.8 and 18.9 Gb for *L. cosentinii* and *L. digitatus*, respectively" as this is a result not a method. Do the same of the Illumina section: "generating ~130 and ~124 million fragments for *L. cosentinii* and *L. digitatus*, respectively". Mention these numbers in the results section.

Page 12 Delete "generating 201 and 354 million fragments for *L. cosentinii* and *L. digitatus*, respectively." Mention these numbers in the results section.

Page 12 "RNA-Seq library preparation". Provide details of the genotype of the plants how they were grown and at which developmental age the tissue was collected. For the leaves what is the difference between the big and small leaves? Were they the same developmental stage and just different in size? With leaf stem, do you mean petiole? Stem, root and pod should all be plural as you collected multiple replicates. There also seem to be different sampling strategies for the two species. Was there a specific reason behind this? Why not consistently sample the same manner for both species? Please provide some more context and details for this in the methods section.

Page 13 remove "yielding ~30 million fragments per sample." Move this detail to the results section.

Page 14 "Structural annotation". Why did the authors only use the predicted protein data and annotations from *L. albus* assembly and not the other Old World Lupin species in *L. angustifolius*?

Page 15 large white space between the words "MCScanX" and "with".

Figure 1 It is nice to show the distribution of the reads, but in my opinion this is not a main Figure, but rather a Supplementary Figure.

Figure 2 The Circus plots are too small to read and therefore leads to confusion when reading the Figure legend where small letters a-d are used to explain the different tracks. The inset for Figure 2B with the different repeat elements requires a larger font size, as does the font size of the x-axis. Figure 2C needs larger front size font size for the numbers in the venn diagram. An additional Figure with the Hi-C contact matrix would be valuable for the readers to determine how well the genome was assembled.

Figure 5 Fonts are too small and the circus plots are too small to read and interpret at this size

Figure 6 is missing labels on both axes. It is also unclear why some of the graphs have red vs black bars. What does the colour difference represent?

Figure 7 Font size on the axes is too small to easily read and interpret the images.

Table 1 Fix up formatting of the "Remaining contig total length (bp)", where (bp) is not entirely legible.

Reviewer #3

(Remarks to the Author)

In this paper, Susek and colleagues provide new genomic resources on the legume genus *Lupinus* and report the genome assembly, annotation and characteristics of the two first genomes of wild Old World species *L. cosentinii* ($2n = 32$) and *L. digitatus* ($2n = 36$). Following structural and genetic analyses the authors found that these genomes have similar genomic structures and that both species are tetraploid with different basic chromosome numbers $x = 8$ and $x = 9$, respectively. They compared these genomes to the domesticated close-relative species *L. albus* and found expansion in genes that could be associated with seed size.

This paper recalls the importance of wild species as a genetic reservoir for improving crops in a context of dramatic climate change, and highlights the remarkable complexity and diversity of the lupin genomes. However, the data remain under-exploited, and, in our opinion, additional analyses are needed to shed more light on the evolutionary dynamics of the lupin genomes and to adequately address the complex seed size genetics in lupins.

The scientific question (about seed size) is not very well addressed. In its current form this manuscript does not provide enough support to the conclusion proposed by the authors.

Thus, at it stands, this manuscript can be hardly considered for publication in Nat-Comm.

Major comments

Introduction:

The introduction lacks clear scientific questions, hypotheses and objectives of this manuscript. For instance, although an interesting and well-documented paragraph on the remarkable genomic diversity and complexity of the lupin genomes, shedding light on their evolutionary history is not underlined as an objective of interest.

Consequences of polyploidy during lupin evolution:

This paragraph presents data supporting the tetraploidy of both *L. cosentinii* and *L. digitatus* and suggests their basic chromosome number (8 and 9, respectively), as assessed by k-mer and biallelic SNPs distribution methods. However, regarding the significant genomic resources generated, and the opportunity offered here to investigate the first lupin genomes with the smallest chromosome numbers in the genus, the data remain underexploited, mainly at the intragenomic level. The authors should have performed appropriate analyses to distinguish the two subgenomes and evaluate their molecular and time divergence (e.g. self Blast, dotplots, intragenomic Ks distributions..)

Similarly, interspecific comparisons with other lupines and with the available conserved ancestral legume genome (ALK) should improve and complement detection of both ancient and recent WGD/WGT events within the genistoid lupin genomes..

Moreover, in the first paragraph of the next section "Comparative genomics in lupin species", the results on colinearity (Fig 5), syntenic blocks (Fig 6) Ka/Ks and Ks distributions (Fig 7) are minimally presented (with unreadable labels, minimal and poorly informative legends), not clearly explained and exploited. As mentioned above, no further intra-genomic analysis of the subgenomes was conducted, which limits their interpretation and scope.

These limits deprive us of important information needed to enrich our understanding on the evolutionary dynamics of lupin genomes and karyotypes. It makes hard to discuss more reliably questions related to the ancient and recent duplication events that occurred during the lupin diversification or related to the fascinating fluctuation of their basic chromosomal numbers (incl. on the ancestral basic # of the genistoids), in a well-understood phylogenetic context. Unfortunately, deciphering "the evolutionary pathways and underlying mechanisms" that shaped the modern lupin genomes is left unanswered).

Paragraph "Comparative genomics in lupin species":

The exclusion of *L. angustifolius* from the syntenic and downstream analyses is a major weakness of this study, as *L. angustifolius* represents another reference genome that could reinforce the comparison between wild species and their close domesticated relatives.

The gene families reconstruction and the analysis of expansions lack critical details in the Method section and it is hard to evaluate if proper parameters have been used and thus if the results support the conclusion made by the authors. Among the missing/unclear information, the following points are important to clarify:

- How gene families have been reconstructed? From the OrthoFinder directly or from MScanX populated with OrthoFinder results?

- The OrthoFinder results look under-used in this study and the manuscript could benefit from a more comprehensive analysis and comparison between species.

- Various crucial details are missing regarding the OrthoFinder reconstruction:

Which species have been used? If only *L. albus*, *L. digitatus* and *L. cosentinii* have been used, I would recommend doing it again integrating *L. angustifolius* to improve the orthogroups resolution as OrthoFinder is more accurate when more species are provided. Integration of *L. angustifolius* could also pave the road to more complete analyses such as comparisons between species with different seed size (small vs large), domesticated vs wild...

Which outgroup has been selected?

What were the search methods retained (OrthoFinder proposes several options):

o Homologs search: diamond, diamond_ultra_sens, blast...

o Orthogroup reconstruction: distance matrix (default) or orthogroup alignment and phylogeny (more accurate)?

Which output of OrthoFinder has been used for downstream analysis? Orthogroups.csv (deprecated) or the phylogenetic hierarchical orthogroups (HOG)?

Did the authors provide a species tree to OrthoFinder or did they let OrthoFinder reconstruct the tree? In the second option, did they check that the estimated tree is consistent with lupine phylogeny?

- The question of the outgroup is very important as it will anchor the species tree and CAFE only considers families that are also present in the outgroup to infer expansion/contraction of the gene families.
- Which parameters have been used for CAFE? Did authors tried multiple CAFE runs without among family rate variation (with multiple gamma categories) as recommended by CAFE's creators?
- Authors identified expansions in *L. cosentinii* and *L. digitatus* compared to *L. albus*, what about contractions?
- Every candidate gene family that show expansion/contraction pattern should be validated by targeted phylogenetic analysis (retrieving the homologs by BLAST and then performing maximum-likelihood or Bayesian inference of the gene families) to ensure that the candidate orthogroup on which the expansion/contraction has been calculated is not artefactual. Indeed, as OrthoFinder uses a single inflation parameter to estimate the orthogroups, it may fail to properly reconstruct the gene families leading to inaccurate estimation of expansions/contractions. Thus, the targeted phylogeny is mandatory to confirm the expansion/contraction pattern.
- What is defined as species-specific gene families? Only the orthogroups where a single species is present with at least two genes? Or did the authors also include genes that are species-specific but not duplicated? The latter doesn't appear in the orthogroups file of OrthoFinder but are contained in another output file.
- Regarding the genes "specific" to *L. cosentinii*:
The term "specific" is misleading as they are also present in *L. albus* and thus not specific to *L. digitatus* but did the author check the assembly of *L. digitatus* (and *L. angustifolius*?) to rule out the possibility that these genes are just not annotated in the other genomes?
- Expansions in *L. digitatus* and *L. cosentinii* are supposed to be "remnants of lineage-specific second duplication". Have the authors a hypothesis about these duplications (e.g. segmental, mediated by transposable elements...?)

The overall conclusion that these expansions play a role in the seed size looks speculative and not strongly supported by the data presented here. Indeed, there are no clear evidence that the expanded gene families presented by the authors are effectively involved in the seed size regulation. Thus, it cannot be ruled out that the seed size is regulated by shared/identical gene families that do not show expansions/contractions across the different lupin species investigated here. A good way to untangle this question would be to identify genes that are differentially expressed during seed formation for these different species and then cross-referenced them with phylogenomic data (phylotranscriptomic). The latter point can fall out of the scope of this paper, but the conclusion that seed size is the result of convergent evolution in lupines should be tuned down or removed and alternative hypothesis explored.

Minor comments:

In Introduction:

Previous information on genome size variation (from flow cytometry) is missing.

In the 2nd paragraph, « Hufnagel et al. 2020 » reference is duplicated.

In the 3rd paragraph, the source of cytogenetic data for New World lupins ($2n$ and x) is missing.

In the same paragraph, the expression « ... providing 'some' insight into ... » looks reductive of the previous lupine genomic contributions. We suggest to readjust this or simply to remove this term.

"*L. cosentinii* Guss": Guss is also written "Gus" few lines after. Otherwise, the authority is only required in the first citation of the species.

Seeds are not a flowering plant innovation (it is for all seed plants that include Gymnosperms and Angiosperms), and the evolutionary success of angiosperms is not based only on seeds. I suggest reformulating this sentence.

In Results and discussion:

Basic information on the main plant material is presented to late in the manuscript (Table 5).

The reference of the *L. albus* genome size is missing.

The citation of 'Salman-Minkov et al 2016' looks inadequately used: too general and confusing. May be to be removed or reformulated to be understood.

Methods:

BLAST parameters are missing in the method sections.

Which version of CAFE5 did the authors use?

Functional annotation: how was defined a best blast hit? I guess that filtering by 80% identity and coverage can return more than one hit for a given gene.

Parameters for CAFE/OrthoFinder are missing (detailed in major comments).

Method used to calculate GO terms enrichment is completely missing.

Reviewer #4

(Remarks to the Author)

Version 1:

Reviewer comments:

Reviewer #1

(Remarks to the Author)

My previous comments have been answered.

There are too many tables and figures. I suggest authors to move some low-value tables/figures to the supplementary file. Some figures can be merged into one and the quality of these figures can be improved.

Reviewer #2

(Remarks to the Author)

The authors have addressed most of my previous concerns. There are a few outstanding changes that are required as outlined below:

Page 2 line 54 replace "contest" with "context"

Page 2 line 55 replace "changes" with "change"

Page 5 line 153 "F1 hybrid". Place the number 1 in subscript

Page 6/7 Lines 185-186. Provide a reference for the "recently reported *L. mutabilis* genome"

Page 7 Lines 187-191. The estimated genome sizes for *L. albus* and *L. angustifolius* based on K-mer prediction and flow cytometry were larger than their actual current assemblies. What are the estimated genome sizes for *L. cosentinii* and *L. digitatus* using such methods? Is this similar trend observed? It would be interesting to expand on this here rather than comparing genome assembly sizes to other legumes.

Page 15 Line 430 correct the spelling of "petioles" to "petioles"

Table 1 and Table 2 provide different BUSCO percentages. In Table 1 it was conducted on the whole genome and in Table 2 on the annotated genes. Does this mean that in all four genomes genes are missing from the annotation?

Figures 1 and 6 Make Circos plots larger so you can read the labels in the plots.

Figures 2, 9 and 10 Species names should be in italics

Reviewer #3

(Remarks to the Author)

Susek and colleagues proposed a revised version of their manuscript entitled "The unexplored diversity of rough-seeded lupins provides rich genomic resources and insights into lupin evolution".

In this version, they significantly improved the manuscript especially the introduction and the presentation of results. The removal of the section about the rough-seed-related genes, clearly improves the manuscript as this hypothesis was not supported by any data presented in the original manuscript. The deepening of the analysis of repeated elements, the extension of structural and genetic comparisons of genomes (including *L. angustifolius*), and the effort developed to provide new information to help reflection for understanding the history of the complex evolution of karyotypes in OW lupins and in general, also significantly enhanced the manuscript. The methods section is now much more detailed.

Our comments have been addressed, but there are still some points that require to be considered.

1. In the Main (Introduction): All the last sentence could be removed from the objectives, and moved (or not) to the core introduction or the conclusion if needed.

2. In section devoted to "composition of repetitive sequences":

* Line 193: correct "ab initio"

* Line 212: did you mean "speciation" (rather than specification)?

* Your "general" observation (p.7-8, lines 213-219) regarding your results on the variable proportions of class-I elements among species is not wrong, but is here inappropriately applied to the abundance of "specific families", while your estimates of TE proportions only concerned the order and superfamily levels, not the family level. So please, adjust/reformulate this.

* By the way, it is surprising why the only paper published on "The Repetitive Content in Lupin Genomes" (Springer book 2020) has been neither cited nor exploited in this section.

3. In section "Consequences of polyploidy during lupin evolution"

We find that the description/explication of the diploidization process of *L. cos* and *L. dig* given by the authors in their answer to the comment 6 of reviewer 1 is well written, and should be used to make clearer the writing of this paragraph in the manuscript (lines 339-346), although some critical points still need to be readjusted (including in the associated Fig S3).

Lines 281-289: The authors should be cautious when interpreting the phylogenetic position of the taxa. According to the *Lupinus* phylogenies cited here (and others) and in agreement with your comparative genomic results, it can be inferred that *L. albus* is most likely the (or one of the) closest smooth seeded lupin relatives of the RS lupins (incl. *L. cos* and *L. dig*). However, the latter are not "intermediate" between the two distinct RS and SS groups, but are fully members of the RS lupins resulting from the diversification of their common ancestor (RSCA) which diverged from the ancestral lineage of *L. albus* ~7.5 Ma. Nevertheless, whether or not the common ancestor of both *L. albus* and the RSCA already riched or not $2n=52$ at the time of the divergence remains unknown at our knowledge to date. Thus, it would be preferable to evaluate chromosome changes of *L. cos* and *L. dig* by comparison to the WGT ancestral lupin ($2n=54$; i.e., changes of 22 or 18 chromosomes, respectively) rather than to *L. albus*. Moreover, regarding that the position and relationships of *L. cos* and *L. dig* (that diverged from each other ~4.5 Ma) have not yet been fully resolved in the published phylogenies, and that the RS lupin clade also contain other closely related species with $2n=38$ and 42, it seems premature to state (only based on the datation) that *L. digitatus* would have derived from *L. cos*. Only additional studies and a fully resolved phylogeny of the RS lupins will be able to lighten this question (including the $2n$ value of their common ancestor) and to make robust hypotheses on the evolutionary steps and on the directionality of the chromosome changes (of $2n$ and x) that experienced these taxa.

According to the above comments, we would suggest to the authors to improve their hypothetical scheme of karyotype evolution Fig S3, especially in the part following the divergence of the RSCA from *L. albus*, keeping only the most reliable information and avoiding phylogenetically erroneous or too speculative scenarios (such as e.g., the arrow going from *L. cos* to *L. dig* ...). Please find attached a draft diagram with our suggestions, which you may find useful. Additionally, the supplementary figure S3 should be accompanied by exhaustive captions for a good understanding by the readers.

4. As mentioned by another reviewer, it is highly surprising that despite all the methods/analyses deployed (Hi-C, BioNano...) genome assemblies are not at chromosome-level. Despite this likely does not influence the outcome of the study, it would be interesting that the authors discuss (even shortly) this point as readers will probably have the same interrogation and this will highlight the complexity of working with some species despite the power of the available tool.

Otherwise, we wonder why the stick chart representation of the chromosomes with the colinearity blocks of Figure 5 (in the initial manuscript) was removed from Figure 6 in the revised manuscript. This representation well complements and is more readable and best illustrative than the circos alone. It would be preferable to maintain this stick chart (with improvement of the labels) in the revised version together with the circos, or with the circos as sup figure.

5. Line 352: 2,784 and 2,751 in *L. cosentinii* and *L. albus*, respectively) should be 2,784 and 2,751 in *L. cosentinii* and *L. digitatus*, respectively) if we correctly understood the sentence.

6. L478: authors state that genes with no functional annotation in InterProScan have been discarded. Is that correct? If yes, this is a huge limit to the study and a potential critical issue as we do not know all the possible functions. For example, even the model plant *Arabidopsis thaliana* or *Medicago truncatula* contain genes with no predicted functions.

7. Regarding the CAFE analysis, authors did not clearly answer my previous comment (maybe I wasn't explicit enough). When running, CAFE, will take the species tree along with orthogroups as input files. For all the downstream analyses, CAFE will consider that all genes have to be present at the root of the tree. Thus, it is important to know which species is the most basal in the tree used by authors when running CAFE. In addition, the tree has to be ultrametric (which is not the case of the species tree produced by OrthoFinder), how did the authors made the tree ultrametric (just scaling branches or including divergence time)?

Reviewer #4

(Remarks to the Author)

Version 2:

Reviewer comments:

Reviewer #2

(Remarks to the Author)

The authors have addressed my previous concerns.

Reviewer #3

(Remarks to the Author)

The authors have correctly addressed our comments and requests. However, we have two final requests for authors:

1- Page 7-Line186: In this paragraph, Fig. S3 (on *L. albus* and *L. angustifolius* genomes) is cited without any link or explication with the text. Something is missing here. Please complement or move citation of Fig. S3 where appropriate.

2- In the "data availability" section, authors provide bioproject IDs for sequenced reads but there is no mention if the newly sequenced genomes and their annotations are included in these bioprojects (they are still inaccessible). Both genome assembly AND annotation files should be made publicly available.

Reviewer #4

(Remarks to the Author)

Reviewer #1 (Remarks to the Author):

Karolina and co-authors have assembled the genomes of two wild lupins. Based on this, they analyzed the polyploidy of the two species, the characteristics of these expanded genes, and the possible association with the large seed size of the two species. I see the limit value of the work as a resource, but the overall analysis/quality of the manuscript is low/rough. I have some comments for the authors to improve the manuscript.

1. In the abstract, the authors say that the two wild lupins have a large seed size, reflecting convergent evolution contributing to lupin domestication. Does the domesticated lupin *L. albus* have a much larger seed size? What is the convergent trait that evolved in which species for its domestication?

These aspects have been removed from the abstract because the emphasis has been adjusted in the main text. The abstract has been also rewritten,

2. When presenting the relationships of the various wild lupins, why do the authors divide them into two groups, New World and Old World? This isn't a scientific grouping of biological entities. Is the geographical distribution of lupins related to the activity of Europeans?

*There are two geographically separated groups in the genus *Lupinus* known as New World and Old World lupins. These groups represent two centres of species diversification and the nomenclature is widely used in the literature (Gladstones et al. 1998). We have included an explanation in the main text and have expanded the discussion on this topic to address the reviewer's concerns. Some lupins have been distributed by humans in Europe but the extent is still unclear (e.g., an East-West migration of *L. angustifolius* has been proposed, Mousavi-Derazmahalleh et al. 2018); Lines 90-97.*

3. I see that you talk more about "New World and Old World lupins" in the conclusion section. I suggest that the authors move this part from the conclusion to the discussion section after the polyploidy study.

We have restructured the discussion/conclusion as suggested; L297-308.

4. Since there are 16 pairs ($2n=32$) and 18 pairs ($2n=38$) of chromosomes for *L. cosentinii* and *L. digitatus* respectively, why did the authors obtain 19 and 22 large scaffolds after two rounds of scaffolding?

*Supplementary Table S2 has been updated with the statistics of the intermediate assemblies to show how each of the two technologies applied A new the scaffolding steps influenced the final assemblies. These assemblies are also affected by the complexity of the species, which is confirmed also by the BUSCO analysis applied to the two published genomes (*L. albus* and *L. angustifolius*) as shown in Table 1.*

5. I'm a bit surprised that simple repeats are the largest classes (22.5% and 15.5%) of repetitive sequences, what kind of simple repeats were extensively amplified, how about these in other closely related species?

*We have repeated the analysis with the same outcome (simple repeats represent ~22.4% in *L. cosentinii* and 15.5% in *L. digitatus*) and have provided more details in the revised manuscript. Importantly, the simple repeats are most abundant only in *L. cosentinii* (where the LTR content is similar, at 21.3%), whereas LTRs are more abundant in *L. digitatus* (20.1%). We found that simple repeats are more abundant in both new assemblies compared to Old World lupin crops (*L. albus* and*

L. angustifolius), whereas LTRs show a contrasting profile. We therefore reported the repeat content and distribution in more detail and made the following adjustments:

- 1) Figure 2a was revised to add the *L. angustifolius* repetitive DNA content, improving the visualization.
- 2) A new Figure 3 was added, representing the evolutionary landscape of DNA transposons and LTR elements among lupin species. This showed that the repetitive DNA content of the two assembled genomes demonstrates a similar evolutionary scenario, while both *L. albus* and *L. angustifolius* have more divergent landscapes. The differences this analysis highlights between *L. cosentinii/L. digitatus* and *L. angustifolius* could suggest that their repetitive DNA content has changed as those species diverged. Figure 2b, Figure 3 and the text have been modified in the results and discussion section 'Gene structure and composition of repetitive sequences'.
- 3) The presence of abundant simple repeats may reflect the substantial fragmentation of the final genomes, which made it more difficult to order the contigs and anchor them to scaffolds. This issue in the reconstruction of highly repetitive regions has influenced their annotation because many of them were fragmented, making them more likely to be identified as simple repeats rather than other superclasses.

6. The authors assume that the basic chromosome number for *L. cosentinii* and *L. digitatus* is $x=8$ and $x=9$, respectively. What is the method and process of deduction? Are the tetraploidization events ancient or recent, i.e. when did they occur (if the two parental subgenomes rearranged after tetraploidization and became diploid again)? Is it a common tetraploidization event shared by the two wild lupins before they diverged, or did it occur independently after they diverged? Is it an auto- or allo-tetraploidization event?

We assumed the basic chromosome number based on our analyses indicating the species are tetraploids. In addition, given there is literature stating that the genistoid clade and lupin diploid ancestor had $x = 9$ chromosomes (Canon et al. 2015, Xu et al. 2020), we deduced that *L. digitatus* ($2n = 36$) possesses two diploid subgenomes, indicating a basic chromosome number of $x = 9$, whereas *L. cosentinii* ($2n = 32$) might have a basic chromosome number of $x = 8$. However, the basic chromosome number might be $x = 9$ for both species, if a progressive rediploidization process has shaped the current genomes of *L. cosentinii* and *L. digitatus*. We hypothesized that rediploidization might have occurred during the evolutionary period that separated *L. cosentinii* and *L. digitatus* from *L. albus*, involving the changes of 18 or 14 chromosomes, respectively, in comparison to *L. albus*. We have added a new figure to expand on this hypothesis (Supplementary Figure S3), which is supported by the fact that *L. cosentinii* and *L. digitatus* have the smallest chromosome number in the *Lupinus* genus, indicating the pressure on plants to reduce their chromosome number. At this stage, we have identified two diploid subgenomes but it was not possible to verify whether autotetraploidization or allotetraploidization events have shaped these genomes; L269-286.

7. The authors state that they have performed a genomic synteny analysis between *L. cosentinii/L. digitatus* and *L. albus*. As mentioned, *L. albus* experienced a WGT event, I'm concerned about how these 3:2 genomic relationships were compared (3 means three subgenomes generated by WGT in *albus*, 2 means two subgenomes generated by WGD in *cosentinii digitatus*)?

The diploid ancestral genome of *L. albus* probably had nine chromosomes (Xu et al. 2020). *L. albus* has three subgenomes and has undergone one round of WGT, shared with *L. angustifolius* (Xu et al.

2020). We cannot exclude the possibility that *L. cosentini* and *L. digitatus* underwent independent WGD events (Supplementary Figure 3) but our analyses indicated that *L. cosentini* and *L. digitatus* have two subgenomes. Based on synteny, most *L. cosentini*/*L. digitatus* scaffolds are mapped 1:1, indicating the same level of collinearity of the genomes. While comparing these species with *L. albus* and *L. angustifolius* chromosomes, we found some scaffolds that mapped to two chromosomes of *L. albus* and *L. angustifolius* (Figures 8). However, dot plot analysis indicated triplication in *L. cosentini* and *L. digitatus* regions (figures below).

8. For the part of the expanded genes associated with the trait, and the GO analysis. I suggest to largely remove them, as they just do not make sense, providing some real features that can be obtained by genomic comparative analysis is much better.

We have removed this part as suggested.

9. There are two similar citations in the second paragraph of the introduction (Hufnagel et al. 2020) (Hufnagel et al. 2020), please correct.

*The double citation has been deleted. We have also cited papers providing the whole-genome sequence of *L. albus* (Xu et al. 2020) and the pangenome of *L. angustifolius* (Garg et al. 2022); L81.*

10. The last sentence in the last paragraph of the Introduction is very abrupt. I suggest that the authors briefly introduce the basic study and results before this sentence.

Much of the introduction has been extensively rewritten to accommodate requests from other reviewers, including the last paragraph. We have limited the revision of the last paragraph (P3L74-P4L113) to adding information on the experimental approach and how it advances the state of the art.

11. Ks distributions estimate the time of sequence/species divergence, does this mean that LdLc are much closer evolutionarily?

*Having included *L. angustifolius* in the analysis, the K_s distribution of the LdLc relationship shows a lower density than the others, indicating a closer relationship. The results and discussion section 'Comparative genomics in lupin species' has been revised along with Figure 8 to explain how the K_s distribution gives insight into the relationship between those two species.*

Reviewer #2 (Remarks to the Author):

The manuscript by Susek and colleagues describes the genome sequencing, assembly and annotation of two Old World Lupin species in *Lupinus cosentinii* and *L. digitatus*, bringing the total number of Old World Lupin genome assemblies to 4 (*L. albus* and *L. angustifolius* are the other two species with chromosome level assemblies). The authors determine that *Lupinus cosentinii* and *L. digitatus* are tetraploid and subsequently conduct some comparisons to the *L. albus* assembly only. The latter seems odd given that there are two quality genome assemblies and annotations available to conduct these comparisons to. I encourage the authors to expand these comparisons to *L. angustifolius*.

L. angustifolius has been included in the work to improve the genome annotation of the two assemblies and to enrich the genomic comparison. Statistical information about the *L. angustifolius* genome has been added to Table 1 and in the results and discussion section ‘First rough-seeded lupin whole-genome sequences: de novo genome assembly’; L166-191.

The gene and repetitive DNA content of those two genomes have been added to Table 2 in the results section ‘Gene structure and composition of repetitive sequences’; L193-253.

L. angustifolius was compared at the gene level with the other three lupin species using Orthofinder to explore the relationships between the four species. Figures 9 and 10 in the results section ‘Comparative genomics in lupin species’ provide more insight about their differences. The text was also modified according to these new results; L328-355.

The authors should also include line numbers to make the review and feedback on the manuscript easier.

Line numbers have been added to the revised manuscript.

Whilst the manuscript is technically sound, more detail in the methodology is required and the quality of the Figures could improve as per my comments below. Furthermore, some results are described in the methodology which should be moved to the results. I congratulate the authors on the generation of two assemblies for these two Old World Lupin species.

Specific comments:

Page 3 Lines 6-8 Define what you mean by wild relatives. This is important as the subsequent sentence suggests you refer to secondary or tertiary gene pools of a genus rather than wild, undomesticated accessions for a species.

We define crop wild relatives (CWRs) as “wild plant species that are taxonomically related to domesticated crops, including the primary, secondary, and tertiary gene pools, as well as weedy plants. They represent a vital reservoir of genetic diversity, which can be utilized through traditional breeding, neodomestication, or genomic tools to enhance crop resilience, productivity, and adaptation to environmental challenges. Beyond supporting the improvement of major crops, CWRs also play a crucial role in enhancing orphan crops—underutilized or neglected species of local importance—thereby contributing to global food security and agricultural sustainability.” We have added an abbreviated version of this definition in the introduction; L68-70.

Page 3 lines 8-11. There are whole genome sequences available for wild relatives of chickpea please expand this section to ensure it is accurate and cite the wild Cicer publications.

We have included the two citations for chickpea: Khan et al. 2024 and Varshney et al. 2024'; L78-79.

Page 3 lines 12-15. There is also a second albus lupin assembly. The Hufnagel assembly is cited twice. Please correct and provide the correct reference for the other assembly.

*The double citation has been deleted. We have also cited papers providing the whole-genome sequence of *L. albus* (Xu et al. 2020) and the pangenome of *L. angustifolius* (Garg et al. 2022); L81.*

Page 3 lines 21-25 It would be good to clarify which traits you refer to when it comes to the domestication of a lupin species and references supporting the level of domestication should be provided.

We have updated the manuscript accordingly; 101-106.

Page 5 de novo assemblies. Why was HiCanu selected as the preferred assembly tool? Were other genome assembly packages suitable for polyploid genomes assessed (e.g. hifiasm, ALLHiC, and PolyGembler)?

*We have repeated our analysis using a de novo assembly prepared with HiFiasm. The resulting genomes show statistics close to the ones obtained with HiCanu, both in dimension and contiguity (N50). BUSCO also reported similar percentages of completeness and duplication. The assemblies are therefore affected by the complexity of the species, which is confirmed also by the BUSCO analysis applied to the two published genomes (*L. albus* and *L. angustifolius*) as shown in Table 1 in the results section 'First rough-seeded lupin whole-genome sequences: de novo genome assembly'; L166-191.*

Page 5 de novo assemblies. With all the data generated one would have thought to be able to scaffold the assembly to level that matches the number of chromosomes. For both the *L. albus* and *L. angustifolius* genomes using a combination of long and short read data coupled with either BioNano or Hi-C data were able to achieve this.

*We fully agree with this comment, but despite our long experience in complex plant genome assembly and the availability of such a large amount of data and their different typologies we have been unable to prepare assemblies better reflecting the chromosome number of these species. Supplementary Table S2 has been updated with the statistics of the intermediate assemblies, showing how each of the technologies applied during the scaffolding steps influenced the final assemblies. These assemblies are also affected by the complexity of the species, which was confirmed by BUSCO analysis applied to the two published genomes (*L. albus* and *L. angustifolius*) as shown in Table 1; L166-191.*

A Hi-C contact map and BioNano maps should be presented.

Bionano Maps of both species have already been uploaded as Supplementary Files with codes SUPPF_0000005572 and SUPPF_0000005571. Hi-C matrices have been added as Supplementary Figure S2.

The authors should compare the statistics of the assemblies to the current *L. albus* and *L. angustifolius* genome statistics. The same goes for the gene content and repetitive sequences. Simply only comparing to albus lupin is not sufficient.

**L. angustifolius* has now been included – please see our response to the first comment.*

Page 7 Comparative genomics. Why only compare to *Lupinus albus* and not both sequenced lupin species? As they are all Old World species it seems strange to omit one species in the comparisons.

*We have added *L. angustifolius* and extended our results to all six pairwise comparisons. The text and the figures/tables have been updated in the results section ‘Comparative genomics in lupin species’; L328-355.*

General comment on the methods section: consistently provide details of manufacturers. The authors currently provide manufacturer details for some but not other reagents/instruments used. For example, no supplier provided for Bionano, TapeStation or AMPureXP beads.

We have added suppliers/manufacturers to all major instruments and reagents but only on first mention, not if the same instrument/reagent is described in subsequent sections of the methods.

Page 10 “Plant materials” Provide details of the cultivar/accessions selected as the representatives of *L. cosentinii* and *L. digitatus* for which you sequenced the genomes and how /from where were these obtained?

*Key details were presented in Table 5 in the original manuscript. We have now incorporated the characterization of the accessions in the main text. We selected *L. cosentinii* 98460 among several accessions based on seed production to secure the proper number of seeds for further research and multiplication. Accession *L. cosentinii* 98460, CV population, country of collection Morocco, was obtained from the Polish *Lupinus* Genebank, Wiatrowo, Poland. We used the only *L. digitatus* accession provided by US Department of Agriculture (ID: PI 660697, collected in Spain). For both species we multiplied SSD lines, considering the conservation of genetic resources; L371-380.*

Page 11 Lines 18-19. Delete “generating ~32.8 and 18.9 Gb for *L. cosentinii* and *L. digitatus*, respectively” as this is a result not a method. Do the same of the Illumina section: “generating ~130 and ~124 million fragments for *L. cosentinii* and *L. digitatus*, respectively”. Mention these numbers in the results section. Page 12 Delete “generating 201 and 354 million fragments for *L. cosentinii* and *L. digitatus*, respectively.” Mention these numbers in the results section.

This information has been moved to the results as requested.

Page 12 “RNA-Seq library preparation”. Provide details of the genotype of the plants how they were grown and at which developmental age the tissue was collected. For the leaves what is the difference between the big and small leaves? Were they the same developmental stage and just different in size? With leaf stem, do you mean petiole? Stem, root and pod should all be plural as you collected multiple replicates.

*The characteristics of *L. cosentinii* and *L. digitatus* are summarized in Table 5. The source material was used to develop single-seed descent lines (at least two rounds of multiplication). Seeds of both species were scarified, vernalized for 21 days and then sown in 7.5-L pots containing a 1:1 mix of peat and vermiculite. The plants were grown in a phytotron at 22/18 °C (day/night temperature) with a 16-h photoperiod, 60–65% relative humidity, and watering as required. This information was already presented in the methods section, but we have now added the IDs of the lupins. We have changed “leaf stem” to “petioles” for clarity. We have also changed the stem, root and pod from singular to plural. Leaves were collected at the same stage for both species. “Small” referred to young leaves (2 weeks old) and “Big” referred to fully developed leaves (5 weeks old) collected from the same plants. We have updated the text to clarify. This information is presented in the methods section; L371-380 and L422-425.*

There also seem to be different sampling strategies for the two species. Was there a specific reason behind this? Why not consistently sample the same manner for both species? Please provide some more context and details for this in the methods section.

*The samples were collected in the same manner at the same stage. The difference is that we collected more tissue for *L. digitatus* because this species was sequenced first. Later we decided to pool young and fully developed leaf tissues as well as lateral and main roots.*

Page 13 remove “yielding ~30 million fragments per sample.” Move this detail to the results section.

We have moved the information to the results as requested.

Page 14 “Structural annotation”. Why did the authors only use the predicted protein data and annotations from *L. albus* assembly and not the other Old World Lupin species in *L. angustifolius*?

*Genome annotation has been repeated, including as extrinsic data the proteomes of both *L. albus* and *L. angustifolius*, as suggested. The proteomic data were also considered during functional annotation at a medium level of confidence, running BLAST analysis on both those datasets to improve the functional description and the GO assignment. The text in the methods section was rewritten accordingly; L463-479.*

Page 15 large white space between the words “MCScanX” and “with”. Figure 1 It is nice to show the distribution of the reads, but in my opinion this is not a main Figure, but rather a Supplementary Figure.

We have corrected the spacing error and moved the distribution data to Supplementary Figure S1.

Figure 2 The Circus plots are too small to read and therefore leads to confusion when reading the Figure legend where small letters a-d are used to explain the different tracks. The inset for Figure 2B with the different repeat elements requires a larger font size, as does the font size of the x-axis. Figure 2C needs larger front size font size for the numbers in the venn diagram.

All the figures and tables have been revised to improve visual clarity.

An additional Figure with the Hi-C contact matrix would be valuable for the readers to determine how well the genome was assembled.

The Hi-C contact matrices have been included as Supplementary Figure S2

Figure 5 Fonts are too small and the circus plots are too small to read and interpret at this size

Figure 6 is missing labels on both axes. It is also unclear why some of the graphs have red vs black bars. What does the colour difference represent?

Figure 7 Font size on the axes is too small to easily read and interpret the images.

All the figures and tables have been revised to improve visual clarity.

Table 1 Fix up formatting of the “Remaining contig total length (bp)”, where (bp) is not entirely legible.

Table 1 has been reformatted.

Reviewer #3 (Remarks to the Author):

In this paper, Susek and colleagues provide new genomic resources on the legume genus *Lupinus* and report the genome assembly, annotation and characteristics of the two first genomes of wild Old World species *L. cosentinii* ($2n = 32$) and *L. digitatus* ($2n = 36$). Following structural and genetic analyses the authors found that these genomes have similar genomic structures and that both species are tetraploid with different basic chromosome numbers $x = 8$ and $x = 9$, respectively. They compared these genomes to the domesticated close-relative species *L. albus* and found expansion in genes that could be associated with seed size.

This paper recalls the importance of wild species as a genetic reservoir for improving crops in a context of dramatic climate change, and highlights the remarkable complexity and diversity of the lupin genomes. However, the data remain under-exploited, and, in our opinion, additional analyses are needed to shed more light on the evolutionary dynamics of the lupin genomes and to adequately address the complex seed size genetics in lupins.

The scientific question (about seed size) is not very well addressed. In its current form this manuscript does not provide enough support to the conclusion proposed by the authors. Thus, at it stands, this manuscript can be hardly considered for publication in Nat-Comm.

Major comments

Introduction: The introduction lacks clear scientific questions, hypotheses and objectives of this manuscript. For instance, although an interesting and well-documented paragraph on the remarkable genomic diversity and complexity of the lupin genomes, shedding light on their evolutionary history is not underlined as an objective of interest.

The introduction has been extensively revised to address the reviewer's comments.

Consequences of polyploidy during lupin evolution: This paragraph presents data supporting the tetraploidy of both *L. cosentinii* and *L. digitatus* and suggests their basic chromosome number (8 and 9, respectively), as assessed by k-mer and biallelic SNPs distribution methods. However, regarding the significant genomic resources generated, and the opportunity offered here to investigate the first lupin genomes with the smallest chromosome numbers in the genus, the data remain underexploited, mainly at the intragenomic level. The authors should have performed appropriate analyses to distinguish the two subgenomes and evaluate their molecular and time divergence (e.g. self Blast, dotplots, intragenomic Ks distributions..)

We have now included the Ks distribution (Figure 8) and dot plot ((figures below) as suggested, but we have omitted the self-Blast because the dot plot already represents the self-alignment of the genome.

Similarly, interspecific comparisons with other lupines and with the available conserved ancestral legume genome (ALK) should improve and complement detection of both ancient and recent WGD/WGT events within the genistoid lupin genomes

We provided more details about our analyses (mainly OrthoFinder). We are fully aware of the existing ancestral legume karyotype ALK with $n = 16$ (Hufnagel et al. 2020), but the number of chromosomes in the legume ancestral tetraploid remains unclear, and could be for example $n = 9$ (Cannon et al. 2019) or $n = 11$ (Wang et al. 2017), this we decided not to it as a reference.

Moreover, in the first paragraph of the next section “Comparative genomics in lupin species”, the results on colinearity (Fig 5), syntenic blocks (Fig 6) Ka/Ks and Ks distributions (Fig 7) are

minimally presented (with unreadable labels, minimal and poorly informative legends), not clearly explained and exploited.

All the figures and tables have been revised to improve visual clarity. The text of this paragraph has also been rewritten to explain the results in more detail.

As mentioned above, no further intra-genomic analysis of the subgenomes was conducted, which limits their interpretation and scope. These limits deprive us of important information needed to enrich our understanding on the evolutionary dynamics of lupin genomes and karyotypes. It makes hard to discuss more reliably questions related to the ancient and recent duplication events that occurred during the lupin diversification or related to the fascinating fluctuation of their basic chromosomal numbers (incl. on the ancestral basic # of the genistoids), in a well-understood phylogenetic context. Unfortunately, deciphering “the evolutionary pathways and underlying mechanisms” that shaped the modern lupin genomes is left unanswered).

The intention of the study was to compare two rough-seeded lupin species with two smooth-seeded species rather than the subgenomes within each species. However, we have now included some intra-genomic analysis (please see response to previous comment).

Paragraph “Comparative genomics in lupin species”: The exclusion of *L. angustifolius* from the syntenic and downstream analyses is a major weakness of this study, as *L. angustifolius* represents another reference genome that could reinforce the comparison between wild species and their close domesticated relatives.

*We have now included *L. angustifolius* and have repeated the comparative analysis involving all four species. The revised data are presented in Figures 8 and 9 and in the results paragraph ‘Comparative genomics in lupin species’; L328-355.*

The gene families reconstruction and the analysis of expansions lack critical details in the Method section and it is hard to evaluate if proper parameters have been used and thus if the results support the conclusion made by the authors. Among the missing/unclear information, the following points are important to clarify:

• How gene families have been reconstructed? From the OrthoFinder directly or from MScanX populated with OrthoFinder results?

*The gene families were obtained from the Orthofinder analysis, which has been run again to include *L. angustifolius* in the comparison. MScanX was used only to determine syntenic relationships, but we considered as gene families the orthogroups obtained in the Orthofinder output, specifically, the N0.tsv file in phylogenetic hierarchical orthogroups; L509-522.*

• The OrthoFinder results look under-used in this study and the manuscript could benefit from a more comprehensive analysis and comparison between species.

We have re-analyzed the OrthoFinder results to better evaluate the orthogroups/gene families shared by the four genomes and the private content of each genome (Figure 9). Also, a phylogenetic tree constructed in Orthofinder using the STAG algorithm based on the four species was added as Figure 10 and the results have been updated to reflect the new data; L345-L355; L509-522.

• Various crucial details are missing regarding the OrthoFinder reconstruction:

- **Which species have been used? If only *L. albus*, *L. digitatus* and *L. cosentinii* have been used, I would recommend doing it again integrating *L. angustifolius* to improve the orthogroups resolution as OrthoFinder is more accurate when more species are provided. Integration of *L. angustifolius* could also pave the road to more complete analyses such as comparisons between species with different seed size (small vs large), domesticated vs wild...**

*As recommended, *L. angustifolius* has now been included to identify the relationships between the four species. More insights about their differences are provided in Figures 8 and 9 and the results section ‘Comparative genomics in lupin species’ has been updated to reflect this. These results set the scene for our discussion of the processes underlying the evolution of smooth- and rough-seeded species; L328-355.*

- **Which outgroup has been selected?**

Both OrthoFinder and CAFE were run with all four species, and the contracted/expanded families and their GO enrichment were determined by comparing each of the assembled genomes against one of the two published genomes, independently. So, we have evaluated the loss/gain of genes in the two assembled genomes concerning one of the two published lupins, separately. We did not use any outgroup because we preferred to choose related species that were well assembled and well annotated. The phylogenetic tree is indeed in line with the expected Lupins hierarchy (Drummond et al. 2012). Introducing an outgroup with our assemblies would have led to possible overprediction in the annotation. The method section ‘Comparative genomics’ has been modified accordingly; L509-529.

- **What were the search methods retained (OrthoFinder proposes several options):**
 - o **Homologs search: diamond, diamond_ultra_sens, blast...**

We ran OrthoFinder with default parameters, including a homologous search with diamond. This is now reported in the methods section; L509-510.

- **Orthogroup reconstruction: distance matrix (default) or orthogroup alignment and phylogeny (more accurate)?**

*Orthogroup reconstruction was achieved using a distance matrix. We also ran Orthofinder with the parameter *-M msa* for orthogroup alignment and phylogeny, but observed no differences.*

- **Which output of OrthoFinder has been used for downstream analysis? Orthogroups.csv (deprecated) or the phylogenetic hierarchical orthogroups (HOG)?**

*The downstream analysis of OrthoFinder output used the *N0.tsv* file, the output in the phylogenetic hierarchical orthogroups, as the software’s GitHub page suggests. This is now reported in the methods section; L511-513.*

- **Did the authors provide a species tree to OrthoFinder or did they let OrthoFinder reconstruct the tree? In the second option, did they check that the estimated tree is consistent with lupine phylogeny?**

*The phylogenetic tree constructed by Orthofinder with the STAG algorithm was based on the four species as reported in Figure 10. Its structure reflects the relationship known so far about *Lupinus* species (e.g., Drummond et al. 2012)*

- **The question of the outgroup is very important as it will anchor the species tree and CAFE only considers families that are also present in the outgroup to infer expansion/contraction of the gene families.**

As mentioned above, we did not use any outgroup because we preferred to choose related species that were well assembled and well annotated. The phylogenetic tree is indeed in line with the expected Lupinus hierarchy. Introducing an outgroup with our assemblies would have led to possible overprediction in the annotation.

- **Which parameters have been used for CAFE? Did authors tried multiple CAFE runs without among family rate variation (with multiple gamma categories) as recommended by CAFE's creators?**

CAFE was run with default parameters initially and then different gamma (-k) were tested. We did not observe relevant differences between the different gamma settings in comparison with the Gene_families_Results.txt from which we started the expansion/contraction analysis. We also contacted the developers who told us to use a high value of 7 for a more accurate result. The text of paragraph 'Comparative genomics' in the methods section was modified accordingly; L523-529.

- **Authors identified expansions in L. cosentinii and L. digitatus compared to L. albus, what about contractions?**

The analysis identified both expansions and contractions (Table 4 and Supplementary Table S6, in the previous version of the manuscript). The text of results section 'Comparative genomics in lupin species' has been updated to include results about the contracted families, but as recommended by another reviewer, we decided to remove this aspect of the study from the manuscript.

- **Every candidate gene family that show expansion/contraction pattern should be validated by targeted phylogenetic analysis (retrieving the homologs by BLAST and then performing maximum-likelihood or Bayesian inference of the gene families) to ensure that the candidate orthogroup on which the expansion/contraction has been calculated is not artefactual. Indeed, as OrthoFinder uses a single inflation parameter to estimate the orthogroups, it may fail to properly reconstruct the gene families leading to inaccurate estimation of expansions/contractions. Thus, the targeted phylogeny is mandatory to confirm the expansion/contraction pattern.**

We have removed this aspect of the manuscript, also because of the recommendation of another reviewer.

- **What is defined as species-specific gene families? Only the orthogroups where a single species is present with at least two genes? Or did the authors also include genes that are species-specific but not duplicated? The latter doesn't appear in the orthogroups file of OrthoFinder but are contained in another output file.**

An orthogroup which contains genes from a single species is considered as a species-specific gene family because those genes are all paralogs. Duplicated genes were also considered. This point has been made clearer in the text in the methods section 'Comparative genomics'; L509-522.

- **Regarding the genes "specific" to L. cosentinii: The term "specific" is misleading as they are also present in L. albus and thus not specific to L. digitatus but did the author check the assembly of L. digitatus (and L. angustifolius?) to rule out the possibility that these genes are just not annotated in the other genomes?**

Given that the L. albus and L. angustifolius proteomes were used as extrinsic evidence for the structural annotation of the genomes, genes private to L. cosentinii or L. digitatus cannot be present in the other two published genomes. Moreover, Orthofinder analysis provided valid insights into the gene families between the species as reported in Figure 9.

• Expansions in L. digitatus and L. cosentinii are supposed to be “remnants of lineage-specific second duplication”. Have the authors a hypothesis about these duplications (e.g. segmental, mediated by transposable elements...?)

The overall conclusion that these expansions play a role in the seed size looks speculative and not strongly supported by the data presented here. Indeed, there are no clear evidence that the expanded gene families presented by the authors are effectively involved in the seed size regulation. Thus, it cannot be ruled out that the seed size is regulated by shared/identical gene families that do not show expansions/contractions across the different lupin species investigated here. A good way to untangle this question would be to identify genes that are differentially expressed during seed formation for these different species and then cross-referenced them with phylogenomic data (phylotranscriptomic). The latter point can fall out of the scope of this paper, but the conclusion that seed size is the result of convergent evolution in lupines should be tuned down or removed and alternative hypothesis explored.

We have removed this aspect of the study from the manuscript.

Minor comments:

In Introduction: Previous information on genome size variation (from flow cytometry) is missing.

This information is included in Table 5.

In the 2nd paragraph, « Hufnagel et al. 2020 » reference is duplicated.

The double citation has been deleted. We have also cited papers providing the whole-genome sequence of L. albus (Xu et al. 2020) and the pangenome of L. angustifolius (Garg et al. 2022); L81.

In the 3rd paragraph, the source of cytogenetic data for New World lupins (2n and x) is missing.

We added the somatic chromosome numbers of:

L. mutabilis (2n=48); L98

L. linearis (2n =32, 34), L. bracteolaris (2n = 32, 34), L. linearis (2n =32, 34) L. cumulicola (2n = 52) and L. villosus; L112-113.

The suggested basic chromosome number in New World lupins is x=6, so we decided not to add these data for each species, but have instead stated that “the basic chromosome number is proposed to be x = 6 in all cases”. The citations have also been provided; L113-115.

In the same paragraph, the expression « ... providing ‘some’ insight into ... » looks reductive of the previous lupine genomic contributions. We suggest to readjust this or simply to remove this term.

We have changed this to a more definitive statement “...providing insight into key aspects of lupin genome structure, diversity and evolution...”; L82-83.

“L. cosentinii Guss”: Guss is also written “Gus” few lines after. Otherwise, the authority is only required in the first citation of the species.

L. cosentinii Guss. is the correct name, and we have deleted the second, erroneous, statement; L143.

Seeds are not a flowering plant innovation (it is for all seed plants that include Gymnosperms and Angiosperms), and the evolutionary success of angiosperms is not based only on seeds. I suggest reformulating this sentence.

The statement has been removed.

In Results and discussion:

Basic information on the main plant material is presented too late in the manuscript (Table 5).

We have added the key information to the main text in the methods section; L371-380.

The reference of the L. albus genome size is missing.

We have added the correct reference to the revised manuscript.

The citation of ‘Salman-Minkov et al 2016’ looks inadequately used: too general and confusing. May be to be removed or reformulated to be understood.

We have removed this citation, as the paragraph has been rewritten.

Methods:

BLAST parameters are missing in the method sections.

BLAST results were filtered considering a minimum identity coverage of 80% and minimum query coverage of 40%. The paragraph ‘De novo genome assembly from PacBio Hi-Fi reads’ in the Methods section has been modified accordingly; L450-451.

Which version of CAFE5 did the authors use?

We used v5.1.0 – this information has been added; L523-525.

Functional annotation: how was defined a best blast hit? I guess that filtering by 80% identity and coverage can return more than one hit for a given gene.

We considered the best BLAST hit as the one with the higher Q-score for each gene and we applied an additional filter of 80% for both identity and query coverage; L481-490.

Parameters for CAFE/OrthoFinder are missing (detailed in major comments). Method used to calculate GO terms enrichment is completely missing.

GO enrichment analysis was implemented with the ‘enricher’ method of the clusterProfiler library, considering only significant results ($p < 0.05$). This information has been added to the methods section; L523-529.

Figures:

A. hypogaea (4) - *M. truncatula* (2) vs. *L. cosentinii*

A. hypogaea (4) vs. *M. truncatula* (2) – *L. digitatus*

Absence of duplicated regions

--- Tripllicated regions

L. japonicus (2) – *P. vulgaris* (2) vs. *M. truncatula* (2)

Absence of duplicated regions

L. japonicus (2) - *M. truncatula* (2) vs. *L. digitatus*

--- Triplicated regions

Dotplots *L. digitatus* - *L. digitatus*

Dotplots *L. cosentinii* - *L. cosentinii*

Dotplots between two assemblies

Effect of polishing on the two assemblies

L.cosentinii

Variants identified in unpolished assembly				Variants identified in polished assembly			
SNP		InDel		SNP		InDel	
Het.	Hom.	Het.	Hom.	Het.	Hom.	Het.	Hom.
57,669	677	10,989	7,474	58,302	541	10,950	279

L.digitatus

Variants identified in unpolished assembly				Variants identified in polished assembly			
SNP		InDel		SNP		InDel	
Het.	Hom.	Het.	Hom.	Het.	Hom.	Het.	Hom.
111,366	2,029	37,808	24,816	156,967	594	45,549	1,692

REVIEWER COMMENTS

Reviewer #1 (Remarks to the Author):

My previous comments have been answered.

There are too many tables and figures. I suggest authors to move some low-value tables/figures to the supplementary file. Some figures can be merged into one and the quality of these figures can be improved.

We have moved Tables 3–5, some parts of Figure 1 and Figure 6, and entire Figure 7 to supplementary data. We have merged Figures 8-10 into one.

Reviewer #2 (Remarks to the Author):

The authors have addressed most of my previous concerns. There are a few outstanding changes that are required as outlined below:

Page 2 line 54 replace “contest” with “context”

Page 2 line 55 replace “changes” with “change”

Page 5 line 153 “F1 hybrid”. Place the number 1 in subscript

We thank the reviewer for highlighting these remaining errors and have corrected them.

Page 6/7 Lines 185-186. Provide a reference for the “recently reported *L. mutabilis* genome

*All references for lupin genomes are provided in the manuscript, but we have added the citation for *L. mutabilis* (and also for other two lupin species) where indicated according to the reviewer’s request. Page 6 Lines 175-177*

Page 7 Lines 187-191. The estimated genome sizes for *L. albus* and *L. angustifolius* based on K-mer prediction and flow cytometry were larger than their actual current assemblies. What are the estimated genome sizes for *L. cosentinii* and *L. digitatus* using such methods? Is this similar trend observed? It would be interesting to expand on this here rather than comparing genome assembly sizes to other legumes.

*The genome sizes of *L. cosentinii* and *L. digitatus*, estimated by flow cytometry, were 695.80 and 671.30 Mbp, respectively (provided in Table 5 of the original manuscript, now moved to Supplementary Table S7). The difference in estimated genome size between flow cytometry and k-mer methods are generally observed in plants, and we have added some discussion on this topic. The table below presents the data obtained based on two different methods. Page 7 Lines 183-191*

		flow cytometry (Mbp/1C)	whole genome assembly (Mbp/1C)	Reference
Lupinus	albus	588	451	Hufnagel et al. 2020
	angustifolius	882	653	Garg et al. 2022
	mutabilis	931	620	Pancaldi et al. 2024
	cosentinii	695.80	588	Susek et al.
	digitatus	671.30	435	Susek et al.

Phaseolus	vulgaris	588	580	Schmutz et al. 2014
Medicago	truncatula	460	430	Pecrix et al. 2018
Lotus	japonicus	480	470	Li et al. 2020

Page 15 Line 430 correct the spelling of “pietoles” to “petioles”

The mistake has been corrected.

Table 1 and Table 2 provide different BUSCO percentages. In Table 1 it was conducted on the whole genome and in Table 2 on the annotated genes. Does this mean that in all four genomes genes are missing from the annotation?

*BUSCO analysis can be run in different modes depending on the input file: genomic, proteomic or transcriptomic. Table 1 presents the BUSCO results for the whole genome (--mode genome), whereas Table 2 shows the results for the protein set, which is the output of the annotation with Augustus and InterProScan (--mode proteins). The BUSCO dataset used, specific to the Fabales order, includes a set of highly conserved genes within this order. Since all four lupins have a completeness of 94–95%, it indicates that ~6% of the Fabales genes are missing in the assembled lupin species (*L. cosentinii*, *L. digitatus*) as well as in the two published species (*L. albus*, *L. angustifolius*). We highlight that our BUSCO analysis of *L. albus* and *L. angustifolius* does not match with the values reported in the original articles because we re-ran the analysis in order to use the same version of BUSCO and the same dataset in each species. So, the final statistics are normalized between the four species.*

Figures 1 and 6 Make Circos plots larger so you can read the labels in the plots.

We have enlarged and sharpened the images as requested. To avoid taking up too much space in the main article, we moved some parts of Figure 1 into the supplementary data - Supplementary Figure S3. We have restored the stick chart from Figure 6 as suggested by Reviewer#3 and we have moved circos plots from Figure 6 to the supplementary information – Supplementary Figures S5-S10.

Figures 2, 9 and 10 Species names should be in italics

The species names have been corrected as requested.

Reviewer #3 (Remarks to the Author):

Susek and colleagues proposed a revised version of their manuscript entitled “The unexplored diversity of rough-seeded lupins provides rich genomic resources and insights into lupin evolution”.

In this version, they significantly improved the manuscript especially the introduction and the presentation of results. The removal of the section about the rough-seed-related genes, clearly improves the manuscript as this hypothesis was not supported by any data presented in the original manuscript. The deepening of the analysis of repeated elements, the extension of structural and genetic comparisons of genomes (including *L. angustifolius*), and the effort developed to provide new information to help reflection for understanding the history of the complex evolution of karyotypes in OW lupins and in

general, also significantly enhanced the manuscript. The methods section is now much more detailed.

Our comments have been addressed, but there are still some points that require to be considered.

1. In the Main (Introduction): All the last sentence could be removed from the objectives, and moved (or not) to the core introduction or the conclusion if needed.

We have moved the last sentence of the introduction to the conclusions as suggested. Page 13 Lines 383-387

2. In section devoted to “composition of repetitive sequences”:

*** Line 193: correct “ab initio”**

*** Line 212: did you mean “speciation” (rather than specification)?**

We have made the requested changes.

*** Your “general” observation (p.7-8, lines 213-219) regarding your results on the variable proportions of class-I elements among species is not wrong, but is here inappropriately applied to the abundance of “specific families”, while your estimates of TE proportions only concerned the order and superfamily levels, not the family level. So please, adjust/reformulate this.**

We have amended the text by adding more accurate name: superfamily. The section “Gene structure and composition of the repetitive sequences” of Results and discussion has also been revised accordingly. Page 8 Line 215 and Line 217

*** By the way, it is surprising why the only paper published on “The Repetitive Content in Lupin Genomes” (Springer book 2020) has been neither cited nor exploited in this section.**

We have now cited the chapter and discussed some of its content in the revised manuscript. Page 7 Line 209, Page 8 Lines 220-223

3. In section “Consequences of polyploidy during lupin evolution” We find that the description/explication of the diploidization process of *L. cos* and *L. dig* given by the authors in heir answer to the comment 6 of reviewer 1 is well written, and should be used to make clearer the writing of this paragraph in the manuscript (lines 339-346), although some critical points still need to be readjusted (including in the associated Fig S3).

Lines 281-289: The authors should be cautious when interpreting the phylogenetic position of the taxa. According to the *Lupinus* phylogenies cited here (and others) and in agreement with your comparative genomic results, it can be inferred that *L. albus* is most likely the (or one of the) closest smooth seeded lupin relatives of the RS lupins (incl. *L. cos* and *L. dig*). However, the latter are not “intermediate” between the two distinct RS and SS groups, but are fully members of the RS lupins resulting from the diversification of their common ancestor (RSCA) which diverged from the ancestral lineage of *L. albus* ~7.5 Ma. Nevertheless, whether or not the common ancestor of both *L. albus* and the RSCA already riched or not $2n= 52$ at the time of the divergence remains unknown at our

knowledge to date. Thus, it would be preferable to evaluate chromosome changes of *L. cos* and *L. dig* by comparison to the WGT ancestral lupin ($2n=54$; i.e., changes of 22 or 18 chromosomes, respectively) rather than to *L. albus*. Moreover, regarding that the position and relationships of *L. cos* and *L. dig* (that diverged from each other ~4.5 Ma) have not yet been fully resolved in the published phylogenies, and that the RS lupin clade also contain other closely related species with $2n=38$ and 42 , it seems premature to state (only based on the datation) that *L. digitatus* would have derived from *L. cos*. Only additional studies and a fully resolved phylogeny of the RS lupins will be able to lighten this question (including the $2n$ value of their common ancestor) and to make robust hypotheses on the evolutionary steps and on the directionality of the chromosome changes (of $2n$ and x) that experienced these taxa. According to the above comments, we would suggest to the authors to improve their hypothetical scheme of karyotype evolution Fig S3, especially in the part following the divergence of the RSCA from *L. albus*, keeping only the most reliable information and avoiding phylogenetically erroneous or too speculative scenari (such as e.g., the arrow going from *L. cos* to *L. dig* ...). Please find attached a draft diagram with our suggestions, which you may find useful. Additionally, the supplementary figure S3 should be accompanied by exhaustive captions for a good understanding by the readers.

We thank the reviewer for this detailed explanation and we have revised the text accordingly as well as the associated figure (Supplementary Figure S4). However, we have kept the chromosome number in L. albus as $2n = 50$, not as suggested $2n = 52$. Page 10 Lines 271-290

4. As mentioned by another reviewer, it is highly surprising that despite all the methods/analyses deployed (Hi-C, BioNano...) genome assemblies are not at chromosome-level. Despite this likely does not influence the outcome of the study, it would be interesting that the authors discuss (even shortly) this point as readers will probably have the same interrogation and this will highlight the complexity of working with some species despite the power of the available tool.

We have added a short discussion explaining why the Hi-C and Bionano methods were unable to produce chromosome level assemblies due to the complexity of these lupin genomes. Page 6 Lines 168-173

Otherwise, we wonder why the stick chart representation of the chromosomes with the colinearity blocks of Figure 5 (in the initial manuscript) was removed from Figure 6 in the revised manuscript. This representation well complements and is more readable and best illustrative than the circos alone. It would be preferable to maintain this stick chart (with improvement of the labels) in the revised version together with the circos, or with the circos as sup figure.

We have restored the stick chart as suggested. In addition, we have moved the Circos plots from Figure 6 to the supplementary information – Supplementary Figures S5-S10.

5. Line 352: 2,784 and 2,751 in *L. cosentinii* and *L. albus*, respectively) should be 2,784 and 2,751 in *L. cosentinii* and *L. digitatus*, respectively) if we correctly understood the sentence.

The reviewer is correct and we have changed the numbers accordingly.

6. L478: authors state that genes with no functional annotation in InterProScan have been discarded. Is that correct? If yes, this is a huge limit to the study and a potential critical issue as we do not know all the possible functions. For example, even the model plant *Arabidopsis thaliana* or *Medicago truncatula* contain genes with no predicted functions.

*InterProScan was used to filter the genes predicted by Augustus to identify those that are structurally related to known protein domains. That does not mean that we would not find genes with unknown functions. Indeed 22.8% of *L. cosentinii* genes and 18.5% of *L. digitatus* genes in the final annotation did not present any function. This may reflect a non-concordant annotation between InterProScan and RefSeq or other databases. We have clarified this point in the section “Structural annotation” of Methods. Page 17 Lines 496-498*

7. Regarding the CAFE analysis, authors did not clearly answer my previous comment (maybe I wasn't explicit enough). When running, CAFE, will take the species tree along with orthogroups as input files. For all the downstream analyses, CAFE will consider that all genes have to be present at the root of the tree. Thus, it is important to know which species is the most basal in the tree used by authors when running CAFE. In addition, the tree has to be ultrametric (which is not the case of the species tree produced by OrthoFinder), how did the authors made the tree ultrametric (just scaling branches or including divergence time)?

*We confirm that the phylogenetic tree used for the CAFE analysis and shown in Figure 10 (currently Figure 7c) is ultrametric and different from that obtained by Orthofinder. However, there is no difference between the two trees regarding the species disposition – the only changes are the ratio of distances between them. From a design perspective there is no difference, and the most basal species is *Lupinus angustifolius*. Regarding the construction of the ultrametric tree, we processed the Othofinder tree using treePL to convert it. We have added this information to the methods section “Comparative genomics”. Page 18 Lines 532-533*

Reviewer #4 (Remarks to the Author):

** See Nature Portfolio's author and referees' website at www.nature.com/authors for information about policies, services and author benefits.

This email has been sent through the Springer Nature Tracking System NY-610A-NPG&MTS

Confidentiality Statement:

This e-mail is confidential and subject to copyright. Any unauthorised use or disclosure of its contents is prohibited. If you have received this email in error please notify our Manuscript Tracking System Helpdesk team at <http://platformsupport.nature.com> .

Details of the confidentiality and pre-publicity policy may be found here <http://www.nature.com/authors/policies/confidentiality.html>

Privacy Policy | Update Profile

Original Figure S3 of Susek et al.

Figure S3: Relationships among four lupin species with hypothetical scheme of karyotype evolution.

Suggestions to the authors to improve their hypothetical scheme of karyotype evolution in *Lupinus*.

Legends:

RSCA: Common ancestor of the OW rough seeded lupins

Δ : chromosome reduction relative to the WGT ancestral lupin ($2n = 54$)

Divergence dates in lupins are from Drummond et al. 2012

*: from this study